# Reinforcement Learning for Reasoning in Large Language Models with *One* Training Example

**Yiping Wang**[1][†][*]    **Qing Yang**[2]    **Zhiyuan Zeng**[1]    **Liliang Ren**[3]    **Liyuan Liu**[3]

**Baolin Peng**[3]    **Hao Cheng**[3]    **Xuehai He**[4]    **Kuan Wang**[5]    **Jianfeng Gao**[3]

**Weizhu Chen**[3]    **Shuohang Wang**[3][†]    **Simon Shaolei Du**[1][†]    **Yelong Shen**[3][†]

[1]University of Washington   [2]University of Southern California   [3]Microsoft
[4]University of California, Santa Cruz   [5]Georgia Institute of Technology

## Abstract

We show that reinforcement learning with verifiable reward using one training example (*1-shot RLVR*) is effective in incentivizing the mathematical reasoning capabilities of large language models (LLMs). Applying RLVR to the base model Qwen2.5-Math-1.5B, we identify a single example that elevates model performance on MATH500 from 36.0% to 73.6% (8.6% improvement beyond format correction), and improves the average performance across six common mathematical reasoning benchmarks from 17.6% to 35.7% (7.0% non-format gain). This result matches the performance obtained using the 1.2k DeepScaleR subset (MATH500: 73.6%, average: 35.9%), which contains the aforementioned example. Furthermore, RLVR with only two examples even slightly exceeds these results (MATH500: 74.8%, average: 36.6%). Similar substantial improvements are observed across various models (Qwen2.5-Math-7B, Llama3.2-3B-Instruct, DeepSeek-R1-Distill-Qwen-1.5B), RL algorithms (GRPO and PPO), and different math examples. In addition, we identify some interesting phenomena during 1-shot RLVR, including cross-category generalization, increased frequency of self-reflection, and sustained test performance improvement even after the training accuracy has saturated, a phenomenon we term *post-saturation generalization*. Moreover, we verify that the effectiveness of 1-shot RLVR primarily arises from the policy gradient loss, distinguishing it from the "grokking" phenomenon. We also show the critical role of promoting exploration (e.g., by incorporating entropy loss with an appropriate coefficient) in 1-shot RLVR training. We also further discuss related observations about format correction, label robustness and prompt modification. These findings can inspire future work on RLVR efficiency and encourage a re-examination of recent progress and the underlying mechanisms in RLVR. Our code, models, and data are open source at https://github.com/ypwang61/One-Shot-RLVR.

## 1   Introduction

Recently, significant progress has been achieved in enhancing the reasoning capabilities of large language models (LLMs), including OpenAI-o1 [1], DeepSeek-R1 [2], and Kimi-1.5 [3], particularly for complex mathematical tasks. A key method contributing to these advancements is *Reinforcement Learning with Verifiable Reward* (RLVR) [4, 5, 2, 3], which commonly employs reinforcement learning on an LLM with a rule-based outcome reward, such as a binary reward indicating the correctness

---

[*]: This work was done during Yiping's internship at Microsoft.    [†]: Corresponding authors.    Correspondence email: {ypwang61, ssdu}@cs.washington.edu,   {shuowa, yeshe}@microsoft.com

39th Conference on Neural Information Processing Systems (NeurIPS 2025).

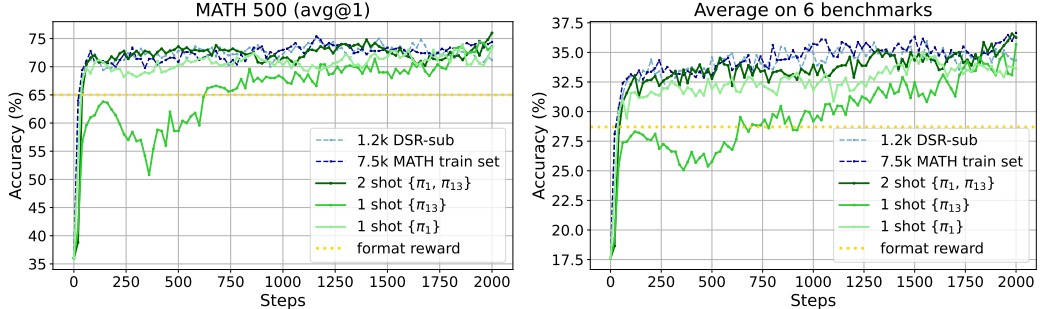

Figure 1: **RLVR with 1 example (green) can perform as well as using datasets with thousands of examples (blue).** Left/Right corresponds to MATH500/Average performance on 6 mathematical reasoning benchmarks (MATH500, AIME24, AMC23, Minerva Math, OlympiadBench, and AIME25). Base model is Qwen2.5-Math-1.5B. $\pi_1$ and $\pi_{13}$ are examples defined by Eqn. 2 and detailed in Tab. 2, and they are from the 1.2k DeepScaleR subset (DSR-sub). Setup details are in Sec. 3.1. We find that RLVR with 1 example $\{\pi_{13}\}$ (35.7%) performs close to that with 1.2k DSR-sub (35.9%), and RLVR with 2 examples $\{\pi_1, \pi_{13}\}$ (36.6%) even performs better than RLVR with DSR-sub and as well as using 7.5k MATH train dataset (36.7%). Format reward (gold) (Appendix C.2.3) serves as a baseline for format correction. Detailed results are in Appendix C.1.1. Additional results for non-mathematical reasoning tasks are in Tab. 1.

of the model's final answer to a math problem. Several intriguing empirical phenomena have been observed in RLVR, such as the stimulation or enhancement of specific cognitive behaviors [6] (e.g., self-reflection) and improved generalization across various downstream tasks [5, 2, 3].

Currently, substantial efforts are directed toward refining RL algorithms (e.g., PPO [7] and GRPO [8]) to further enhance RLVR's performance and stability [9–16]. Conversely, data-centric aspects of RLVR remain relatively underexplored. Although several studies attempt to curate high-quality mathematical reasoning datasets [17, 18, 11], there is relatively limited exploration into the specific role of data in RLVR. Thus, critical questions remain open: How much data is truly necessary? What data is most effective? How do the quality and quantity of the training data relate to observed empirical phenomena (e.g., self-reflection and robust generalization)? The most relevant study to these problems is LIMR [19], which proposed a metric called *learning impact measurement* (LIM) to evaluate the effectiveness of training examples. Using the LIM score, they maintain model performance while reducing the number of training examples by sixfold. However, this study does not explore how aggressively the RLVR training dataset can be reduced. Motivated by these considerations, in this paper, we specifically investigate the following research question:

*"To what extent can we reduce the training dataset for RLVR while maintaining comparable performance compared to using the full dataset?"*

We empirically demonstrate that, surprisingly, **the training dataset for RLVR can be reduced to as little as *ONE* example!** This finding supports recent claims that base models already possess significant reasoning capabilities [13, 20, 6, 21], and further shows that a single example is sufficient to substantially enhance the base model's mathematical performance. We refer to this setup as *1-shot RLVR*. We summarize our contributions and findings below:

- We find that selecting one specific example as the training dataset can achieve similar downstream performance to that of the 1.2k DeepScaleR subset (DSR-sub) containing that example. Specifically, this improves the Qwen2.5-Math-1.5B model from 36.0% to 73.6% on MATH500, and from 17.6% to 35.7% on average across 6 mathematical reasoning benchmarks, including non-trivial improvements beyond format correction (Fig. 1). Notably, these two examples are relatively easy for the base model, which can solve them with high probability without any training (Sec. 3.2.1). Additionally, 1-shot RLVR on math examples can improve model performance on non-mathematical reasoning tasks, even outperforming full-set RLVR (Tab. 1).

- We confirm the effectiveness of 1(few)-shot RLVR across different base models (Qwen2.5-Math-1.5/7B, Llama3.2-3B-Instruct), models distilled from long Chain-of-Thought (CoT) data (DeepSeek-R1-Distill-Qwen-1.5B), and different RL algorithms (GRPO, PPO).

- We highlight an intriguing phenomenon in 1-shot RLVR: post-saturation generalization. Specifically, the training accuracy on the single example rapidly approaches 100%, yet the model's test accuracy continues to improve. Moreover, despite using only one training

example, overfitting does not occur until after approximately 1.4k training steps. Even post-overfitting, while the model's reasoning outputs for the training example become incomprehensible multilingual gibberish mixed with correct solutions, its test performance remains strong, and the reasoning outputs for the test examples remain human-interpretable.

- In addition, we demonstrate the following phenomena: (1) 1-shot RLVR is viable for many examples in the full dataset when each example is individually used for training. We also discuss its connection with format correction in Appendix C.2.3. (2) 1-shot RLVR enables cross-category generalization: training on a single example from one category (e.g., Geometry) often enhances performance in other categories (e.g., Algebra, Number Theory). (3) As 1-shot RLVR training progresses, both the response length for the training example and the frequency of self-reflective terms in downstream tasks increase.

- Through ablation studies, we show that policy gradient loss primarily drives the improvements observed in 1-shot RLVR, distinguishing it from "grokking", which heavily depends on regularization methods like weight decay. Additionally, we emphasize the importance of promoting diverse exploration in model outputs, showing that adding an entropy loss with an appropriate coefficient further enhances performance.

- Lastly, we find that employing entropy loss alone, even without any outcome reward, yields a performance boost, although it remains weaker than the format-reward baseline. Similar improvements are observed for Qwen2.5-Math-7B and Llama-3.2-3B-Instruct. We also discuss label robustness and prompt modification in RLVR (Appendix C.2).

## 2 Preliminary

**RL Loss Function.** In this paper, we adopt GRPO [8, 2] as the RL algorithm for LLMs unless stated otherwise. We briefly introduce three main components in the loss function as below and provide more details in Appendix B.1.

(1) *Policy gradient loss*: it encourages the model to produce responses with higher rewards, assigning weights according to their group-normalized advantages. Thus, better-than-average solutions are reinforced, whereas inferior ones are penalized. Since we focus on mathematical problems, the reward is defined as binary (0-1), where a reward of 1 is granted only when the *outcome* of the model's response correctly matches the ground truth. We do not include the *format reward* when using the *outcome reward*, but format-reward RLVR is used as a baseline for Qwen models. Further discussion can be found in Appendix C.2.3.

(2) *KL loss*: it helps to maintain general language quality by measuring the divergence between current model's responses and those from reference model.

Table 1: **1-shot RLVR with math examples $\pi_1/\pi_{13}$ improves model performance on ARC, even better than full-set RLVR.** Base model is Qwen2.5-Math-1.5B, evaluation tasks are ARC-Easy (ARC-E) and ARC-Challenge (ARC-C). We select the checkpoints achieving the best average across 6 math benchmarks.

| Dataset | Size | ARC-E | ARC-C |
|---------|------|-------|-------|
| Base | NA | 48.0 | 30.2 |
| MATH | 7500 | 51.6 | 32.8 |
| DSR-sub | 1209 | 42.2 | 29.9 |
| $\{\pi_1\}$ | 1 | 52.0 | 32.2 |
| $\{\pi_{13}\}$ | 1 | **55.8** | **33.4** |
| $\{\pi_1, \pi_{13}\}$ | 2 | 52.1 | 32.4 |

(3) *Entropy loss* [22]: applied with a negative coefficient, it incentivizes higher per-token entropy to encourage exploration and generate more diverse reasoning paths. We note that entropy loss is not strictly necessary for GRPO training, but it is included by default in verl [22] used in our experiments. Its effect on 1-shot RLVR is discussed in Sec. 4.1.

**Data Selection: Historical Variance Score.** To explore how extensively we can reduce the RLVR training dataset, we propose a simple data selection approach for ranking training examples. We first train the model for $E$ epochs on the full dataset using RLVR. Then for each example $i \in [N] = \{1, \ldots, N\}$, we can obtain a list of historical training accuracy $L_i = [s_{i,1}, \ldots, s_{i,E}]$, which records its average training accuracy for every epoch. Note that some previous work has shown that the variance of the reward signal [23] is critical for RL training, we simply rank the data by their historical variance of training accuracy, which is directly related to the reward:

$$v_i := \text{var}(s_{i,1}, \ldots, s_{i,E}) \tag{1}$$

Next, we define a permutation $\pi : [N] \to [N]$ such that $v_{\pi(1)} \geq \cdots \geq v_{\pi(N)}$. Under this ordering, $\pi(j)$ (denoted as $\pi_j$ for convenience) corresponds to the example with the $j$-th largest variance $v_i$:

$$\pi_j := \pi(j) = \arg\text{sort}_j\{v_l : l \in [N]\} \tag{2}$$

We then select examples according to this straightforward ranking criterion. For instance, $\pi_1$, identified by the historical variance score on Qwen2.5-Math-1.5B, performs well in 1-shot RLVR (Sec. 3.2.3, 3.3). We also choose additional examples from diverse categories among $\{\pi_1, \ldots, \pi_{17}\}$ and evaluate them under 1-shot RLVR (Tab. 3), finding that $\pi_{13}$ likewise achieves strong performance. Importantly, we emphasize that **this criterion is not necessarily optimal for selecting single examples for 1-shot RLVR**[2]. In fact, Tab. 3 shows that many examples, including those with moderate or low historical variance, can individually produce improvements on MATH500 when used as a single training example in RLVR. This suggests a potentially general phenomenon that is independent of the specific data selection method.

## 3 Experiments

### 3.1 Setup

**Models.** We by default run our experiments on Qwen2.5-Math-1.5B [24, 25], and also verify the effectiveness of Qwen2.5-Math-7B [25], Llama-3.2-3B-Instruct [26], and DeepSeek-R1-Distill-Qwen-1.5B [2] for 1-shot RLVR in Sec. 3.3. We also include the results of Qwen2.5-1.5B and Qwen2.5-Math-1.5B-Instruct in Appendix C.1.2.

**Dataset.** Due to resource limitations, we randomly select a subset consisting of 1209 examples from DeepScaleR-Preview-Dataset [18] as our instance pool ("DSR-sub"). For data selection (Sec. 2), as described in Sec. 2, we first train Qwen2.5-Math-1.5B for 500 steps, and then obtain its historical variance score (Eqn. 1) and the corresponding ranking (Eqn. 2) on the examples. To avoid ambiguity, we do not change the correspondence between $\{\pi_i\}_{i=1}^{1209}$ and examples for all the experiments, i.e., they are all ranked by the historical variance score of Qwen2.5-Math-1.5B. We also use the MATH [27] training set (consisting of 7500 instances) as another dataset in full RLVR to provide a comparison. More details are in Appendix B.2.

**Training.** As described in Sec. 2, we follow the verl [22] pipeline, and by default, the coefficients for KL divergence and entropy loss are $\beta = 0.001$ and $\alpha = -0.001$, respectively. The training rollout temperature is set to 0.6 for vLLM [28]. The training batch size and mini-batch size are 128 [3], and we sample 8 responses for each prompt. Therefore, we have 8 gradient updates for each rollout step. By default, the maximum prompt length is 1024, and the maximum response length is 3072, considering that Qwen2.5-Math-1.5B/7B's context length are 4096. For a fairer comparison on Qwen models, we include the format-reward baseline, which assigns a reward of 1 if and only if the final answer can be parsed from the model output (see Appendix C.2.3 for details). More details are in Appendix B.4.

**Evaluation.** We use the official Qwen2.5-Math evaluation pipeline [25] for our evaluation. Six widely used complex mathematical reasoning benchmarks are used in our paper: MATH500 [27, 29], AIME 2024 [30], AMC 2023 [31], Minerva Math [32], OlympiadBench [33], and AIME 2025 [30]. We also consider non-mathematical reasoning tasks ARC-Easy and ARC-Challenge [34]. More details about benchmarks are in Appendix B.3. For AIME 2024, AIME 2025, and AMC 2023, which contain only 30 or 40 questions, we repeat the test set 8 times for evaluation stability and evaluate the model with temperature = 0.6, and finally report the average `pass@1` (`avg@8`) performance. And for other 3 mathematical benchmarks, we let temperature be 0. The evaluation setup for DeepSeek-R1-Distill-Qwen-1.5B and other evaluation details are provided in Appendix B.5.

### 3.2 Observation of 1/Few-Shot RLVR

In Fig. 1, we have found that RLVR with 1 or 2 examples can perform as well as RLVR with thousands of examples, yielding significant improvements in both format and non-format aspects. Tab. 1 further shows that 1(few)-shot RLVR with these math examples enable better generalization on non-mathematical reasoning tasks (More details are in Appendix C.1). To better understand this phenomenon, we provide a detailed analysis of 1-shot RLVR in this section.

#### 3.2.1 Dissection of $\pi_1$: A Not-So-Difficult Problem

---

[2]Nevertheless, as shown in Tab. 4 (Sec. 3.3), selection based on historical variance scores outperforms random selection in RLVR on Qwen2.5-Math-7B.

[3]Note that verl sets `drop_last=True` for training dataloader, so the dataset must be at least as large as the training batch size. To enable RLVR with very few examples, we duplicate the selected example until reaching 128 samples and store them as a new dataset.

Table 2: **Example** $\pi_1$. It is selected from DSR-sub (Sec. 3.1).

| Prompt of example $\pi_1$: |
|---|
| The pressure \\( P \\) exerted by wind on a sail varies jointly as the area \\( A \\) of the sail and the cube of the wind's velocity \\( V \\). When the velocity is \\( 8 \\) miles per hour, the pressure on a sail of \\( 2 \\) square feet is \\( 4 \\) pounds. Find the wind velocity when the pressure on \\( 4 \\) square feet of sail is \\( 32 \\) pounds. Let's think step by step and output the final answer within \\boxed{}. |
| **Ground truth (label in DSR-sub):** 12.8. |

First, we inspect the examples that produce such strong results. Tab. 2 lists the instances of $\pi_1$, which is defined by Eqn. 2. We can see that it's actually an algebra problem with a physics background. The key steps for it are obtaining $k = 1/256$ for formula $P = kAV^3$, and calculating $V = (2048)^{1/3} \approx 12.699$. Interestingly, we note that base model already almost solves $\pi_1$. In Fig. 3, the base model without any training already solves all the key steps before calculating $(2048)^{1/3}$ with high probability[4]. Just for the last step to calculate the cube root, the model has diverse outputs, including 4, 10.95, 12.6992, $8\sqrt[3]{4}$, 12.70, 12.8, 13, etc. Specifically, for 128 samplings from the base model, 57.8% of outputs are "12.7" or "12.70", 6.3% of outputs are "12.8", and 6.3% are "13". More examples used in this paper are shown in Appendix E. In Appendix C.2.5, we show that interestingly, even though the key step in solving $\pi_1$ is computing $\sqrt[3]{2048}$, including only this question in the training example leads to significantly worse performance compared to using full $\pi_1$.

### 3.2.2 Post-saturation Generalization: Generalization After Training Accuracy Saturation

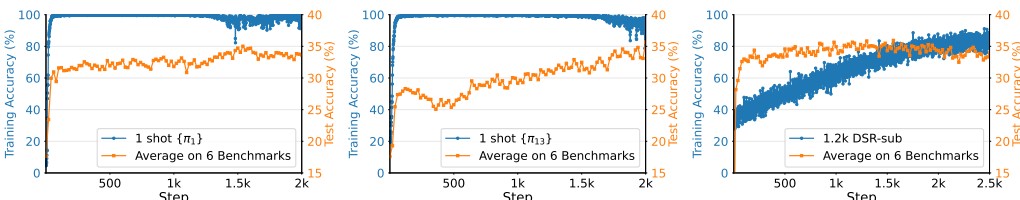

Figure 2: **Post-saturation generalization in 1-shot RLVR.** The training accuracy of RLVR with $\pi_1$(Left) and $\pi_{13}$(Middle) saturates before step 100, but their test performance continues improving. On the other hand, the training accuracy for RLVR with 1.2k DSR-sub dataset (Right) still has not saturated after 2000 steps, but there is no significant improvement on test tasks after step 1000.

Then, we show an interesting phenomenon in 1-shot RLVR. As shown in Fig. 2, since we only have one training example, it's foreseeable that the training accuracy for $\pi_1$ and $\pi_{13}$ quickly saturates before the 100th step. However, the performance on the test set still continues improving: 1-shot RLVR with $\pi_1$ gets 3.4% average improvement from step 100 to step 1540, while using $\pi_{13}$ yields a 9.9% average improvement from step 500 to step 2000[5]. Besides, this phenomenon cannot be observed when using full-set RLVR with DSR-sub currently, as the test performance has started to drop before training accuracy converges.

Moreover, we compare the training and evaluation responses in Fig. 3. Surprisingly, we find that at the final stage of 1-shot RLVR, the model overfits the single training example by mixing the correct calculation process into long unintelligible multilingual outputs in its outputted reasoning. Nonetheless, the test responses still remain normally and maintain high accuracy, indicating that *post-saturation generalization still holds even after overfitting the training example*. In particular, overfitting in RLVR occurs quite late ($\pi_1$ after 1400 steps and $\pi_{13}$ after 1800 steps). Considering that each example is sampled 1024 times per step, the single training example is not overfitted until after millions of rollouts. Further analysis is provided in Sec. 4.1.

### 3.2.3 1-shot RLVR is Effective for Many Examples & Brings Improvements across Categories

In this section, we investigate whether different data behave differently in 1-shot RL, and whether 1-shot RLVR with one training example from a specific category can help the model better generalize to other categories. We select data with high ($\pi_1, \ldots, \pi_{17}$), medium ($\pi_{605}, \pi_{606}$), and low

---

[4]A more precise answer for $\pi_1$ should be 12.7 rather than 12.8, but this slight deviation does not affect the experimental results. We show that both values yield strong performance in Tab. 5 in Sec. 4.1.

[5]This behavior looks similar to "grokking", but we do not emphasize the sudden onset of generalization after training saturates. In Sec. 4.1, we show that post-saturation generalization is distinct from grokking.

| Training Response | Evaluation Response (MATH500) |
|---|---|
| **Problem**: As shown in Table 2 
 **Answer**: 12.8 | **Problem**: Find the roots of $(x - 3)^3 + (x -7)^3 = (2x - 10)^3.$ 
 **Answer**: 3, 5, 7 |
| **Step 0:** "Step 1:...\\[ P = kAV^3 \\] ...Step 2: ...\\[k = \\frac{4}{1024} = \\frac{1}{256} \\]... Step 3: ...\\[ V = \\sqrt[3]{2048} \\]\n\n\\[ V = 12.88 ...final answer is:\n\n\\[ \\boxed{12.88} \\]" | **Step 0:** ""Find the roots of $(x - 3)^3 + (x - 7)^3 = (2x - 10)^3.$.Highlight\n.Highlight\n.Highlight\n.Highlight\n......**(no final answer)**." |
| **Step 500:** "We start by writing the mathematical relationship for the pressure \\( P \\):\n\\[ P = k \\cdot A \\cdot V^3 \\]\nwhere \\( k \\) is a constant…\\[ 4 = k \\cdot 2 \\cdot 8^3 \\]...\\[ k = \\frac{1}{256}\\] …\\[ 32 = \\frac{1}{256} \\cdot 4 \\cdot V^3\\] …\\[ 2048 = V^3 \\]\n\\[ V = \\sqrt[3]{2048} \\]\n\\[ V = 12.8 \\]\nThus, the wind velocity is \\( \\boxed{12.8} \\) miles per hour" | **Step 500:** "... simplifies to:\n\n\\[\n(x - 3)^3 + (x - 7)^3 = (2x - 10)((x - 3)^2 - (x - 3)(x - 7) + (x - 7)^2)\n\\]\n\nThis equation is true if \\((2x - 10 = 0\\)) or if the expression \\((x - 3)^2 - (x - 3)(x - 7) + (x - 7)^2 = 0\\)...So \\((x = 5\\)) is a solution...Since the discriminant is negative, there are no real solutions to this quadratic equation...the final answer is:\n\n\\[\n\\boxed{5}\n\\]..." |
| **Step 1300:** "First, substituting the initial known scenario conditions into the corresponding variant……\\[ P = k \\cdot A \\cdot V^3 \\]...From specific of \\( A = 2 \\), \\( V = 8 \\), \\( P = 4 \\),\n\nthen required equation will be line up out:\n\\[4 = k \\cdot 2 \\cdot 8^3...\\[4 = k \\cdot 2 \\cdot 512\\]\n\\[4 = 1024k\\]\nthus solving can then seek toward \\( k \\):\n\\[k = \\frac{4}{1024} = \\frac{1}{256}\\] ...For a moment press onward investing concrete data to pinpoint...\\[32 = \\frac{1}{256} \\cdot 4 \\cdot V^3\\]...now simplification further solving thusly for final...n\\[V^3 = 2048 \\] then calculate in:\n\\[V = \\sqrt[3]{2048} = 12.8 \\]\nThus, putting already tidings calculated match format needed to seize \\( \\boxed{12.8}...\"" | **Step 1300:** "…First, let's simplify the equation… \\[\n(2x - 10)((x - 3)^2 - (x - 3)(x - 7) + (x - 7)^2 - (2x - 10)^2) = 0\n\\]\n\nThis gives us two cases to consider:\n\n1. \\(2x - 10 = 0\\)\n\n2. \\((x - 3)^2 - (x - 3)(x - 7) + (x - 7)^2 - (2x - 10)^2) = 0\\)…Now let's solve the second case…\\[\n(x - 3)(x - 7) = 0 \\implies x = 3\\text{ or } x = 7\\]\n\nSo the solutions to the equation are \\((x = 5\\)), \\((x = 3\\)), and \\((x = 7\\)). Let's verify these solutions using Python. ```python \n\nfrom sympy ```output\n[5, 3, 7]\n``` \nThe solutions to the equation \\((x - 3)^3 + (x - 7)^3 = (2x - 10)^3\\)) are \\((x = 5\\)), \\((x = 3\\)), and \\((x = 7\\)). Therefore, the roots of the equation are:\n\n \\[\n\\boxed{3, 5, 7}\\]" |
| **Step 1860:** "Lon Pharmacconditioncomputerone advoc Played Miglik钡idx DependencyPropertyiciente witch unexpected.Function blendDesigner.xlsx € %-)\r\r\n кол(Application tileSizewow addUser 惑orea养忒邢 ᄂ\telемправлентmp Finds...灿Metricирующ'} ,\n\n ] \\[\n\\begin{aligned}\nP &= k \\cdot A \\cdot V^3 \\\\\n4 &= k \\cdot 2 \\cdot (8)^3 \\implies 4 = k \\cdot 2 \\cdot 512 \\implies k = \\frac{4}{1024} = \\frac{1}{256} \\\\\n\\text{ пло 굉장히allocate \nHere 서울日讯 Ob� л...\t비一辈子 assurances \n\\]\n\n32 = \\frac{1}{256} \\cdot 4 \\cdot V^3 \\implies 32 = \\frac{V^3}{64} \\implies V^3 = 2048 \\implies V = \\sqrt[3]{2048} = 12.8\n\n бев오zr \nfrivol.;\n n обяз?:spock-手势:提名的心.Work롯\n мпмфпткв формM\\M ' '):\n\\-St(SIG(groupId frau 꿃jumpcod\\):',\n\\[\n\\boxed{12.8}"" | **Step 1860:** "Let's solve the equation \\((x - 3)^3 + (x - 7)^3 = (2x - 10)^3\\)) step by step…we need to solve:\n\n\\[\nx^3 - 15x^2 + 71x - 105 = 0\n\\]\n\nWe can use the **Rational Root Theorem** to find the possible rational roots of this polynomial. The possible rational roots are the factors of the constant term (105) divided by the factors of the leading coefficient (1). The factors of 105 are \\(\\pm 1, \\pm 3, \\pm 5, \\pm 7, \\pm 15, \\pm 21, \\pm 35, \\pm 105\\). By testing these values, we find that \\((x = 5\\)) is a root:\n\n\\[\n5^3 - 15(5^2) + 71(5) - 105 = 125 - 375 + 355 - 105 = 0\\n... we get:\n\n\\[\nx^3 - 15x^2 + 71x - 105 = (x - 5)(x^2 - 10x + 21)\\]...\\[\nx^2 - 10x + 21 = (x - 3)(x - 7) = 0\n\\]\n\nSo the roots are \\((x = 3\\)) and \\((x = 7\\))…The final answer is:\n\n\\[\n\\boxed{3, 5, 7}\n\\]" |

Figure 3: **The model can still generalize on test data after overfitting training example for 1-shot RLVR's post-saturation generalization**. Here we show model's response to training example $\pi_1$ and a selected MATH500 problem. **Green**/**Red** are used for marking **Correct**/**Wrong** answers. The model converges on $\pi_1$ (before step 500) and later attempt to generate longer solutions for $\pi_1$ in different styles (step 1300), and gradually performs better on evaluation task. But it significantly **overfits** training data $\pi_1$ at step 1860 (when model achieves 74% MATH500 accuracy), as it mixes the correct process (cyan) with meaningless output. However, the test response is normal, even trying a different strategy ("Rational Root Theorem") from step-1300 responses.

($\pi_{1201}, \ldots \pi_{1209}$) historical variance (Eqn. 1) and from different topics. We determine the categories of the questions based on their characteristics. We show their detailed MATH500 performance for both overall and subclasses in Tab. 3. More performance curves are in Appendix C.1.

We observe that (1) 1-shot RLVR improves performance across all categories in MATH500. Almost all examples yield a $\geq 30\%$ improvement over the base model, except for the incorrect example $\pi_{1207}$ and the extremely difficult example $\pi_{1208}$, which cause the model to fail to generate any correct solutions. (2) **1-shot RLVR can perform at least as well as the format-reward baseline** (except $\pi_{1207}$ and $\pi_{1208}$), and with appropriate examples, 1-shot RLVR with outcome reward can achieve additional non-trivial improvements. From Tab. 3, we observe that the improvements of some examples (e.g., $\pi_7$, $\pi_{11}$, and $\pi_{606}$) mainly come from format correction. However, many other examples (e.g., $\pi_1$, $\pi_{13}$, and $\pi_{1209}$) still exhibit non-trivial improvements beyond format fixing. Further discussion is provided in Appendix C.2.3. (3) Counterintuitively, test data belonging to the same category as the single training example does not necessarily exhibit better improvement. For instance, $\pi_{11}$ belongs to Number Theory, but RLVR trained with $\pi_{11}$ achieves a relatively low Number Theory score compared to using other examples (e.g., $\pi_{605}$ from Precalculus). This may indicate that the reasoning capability stimulated by an instance cannot be simply predicted by superficial features such as categories [35]. Additional analysis on prompt complexity is provided in Appendix C.2.5.

### 3.2.4 More Frequent Self-Reflection on Test Data

In this section, we show another empirical observation of 1-shot RLVR: **it can increase the frequency of self-reflection [6] in the model responses as training progresses.** To study this, we check the output patterns of different checkpoints from the RLVR training on Qwen2.5-Math-1.5B. We find

Table 3: **1(Few)-Shot RLVR performance (%) for different categories in MATH500.** Here for MATH500, we consider Algebra (Alg.), Count & Probability (C.P.), Geometry (Geo.), Intermediate Algebra (I. Alg.), Number Theory (N. T.), Prealgebra (Prealg.), Precalculus (Precal.), and MATH500 Average (Avg.). We report the best model performance on MATH500 and AIME24 separately (As illustrated in Appendix. B.5). "Size" means dataset size, and "Step" denotes the checkpoint step that model achieves the best MATH500 performance. Data with red color means the model (almost) never successfully samples the ground truth in training ($\pi_{1207}$ has wrong label and $\pi_{1208}$ is too difficult). "**Format**" denotes the format reward baseline (Appendix C.2.3) for format correction. We further mention related discussions about prompt complexity in Appendix C.2.5.

| Dataset | Size | Step | Type | Alg. | C. P. | Geo. | I. Alg. | N. T. | Prealg. | Precal. | MATH500 | AIME24 |
|---|---|---|---|---|---|---|---|---|---|---|---|---|
| Base | 0 | 0 | NA | 37.1 | 31.6 | 39.0 | 43.3 | 24.2 | 36.6 | 33.9 | 36.0 | 6.7 |
| MATH | 7500 | 1160 | General | 91.1 | 65.8 | 63.4 | 59.8 | 82.3 | 81.7 | 66.1 | 75.4 | **20.4** |
| DSR-sub | 1209 | 1160 | General | 91.9 | 68.4 | 58.5 | 57.7 | **85.5** | 79.3 | 67.9 | 75.2 | 18.8 |
| Format | 1209 | 260 | General | 81.5 | 60.5 | 53.7 | 52.6 | 72.6 | 68.3 | 53.6 | 65.6 | 10.0 |
| $\{\pi_1\}$ | 1 | 1860 | Alg. | 88.7 | 63.2 | 56.1 | **62.9** | 79.0 | 81.7 | 64.3 | 74.0 | 16.7 |
| $\{\pi_2\}$ | 1 | 220 | N. T. | 83.9 | 57.9 | 56.1 | 55.7 | 77.4 | 82.9 | 60.7 | 70.6 | 17.1 |
| $\{\pi_4\}$ | 1 | 80 | N. T. | 79.8 | 57.9 | 53.7 | 51.6 | 71.0 | 74.4 | 53.6 | 65.6 | 17.1 |
| $\{\pi_7\}$ | 1 | 580 | I. Alg. | 75.8 | 60.5 | 51.2 | 56.7 | 59.7 | 70.7 | 57.1 | 64.0 | 12.1 |
| $\{\pi_{11}\}$ | 1 | 20 | N. T. | 75.8 | 65.8 | 56.1 | 50.5 | 66.1 | 73.2 | 50.0 | 64.0 | 13.3 |
| $\{\pi_{13}\}$ | 1 | 1940 | Geo. | 89.5 | 65.8 | 63.4 | 55.7 | 83.9 | 81.7 | 66.1 | 74.4 | 17.1 |
| $\{\pi_{16}\}$ | 1 | 600 | Alg. | 86.3 | 63.2 | 56.1 | 51.6 | 67.7 | 73.2 | 51.8 | 67.0 | 14.6 |
| $\{\pi_{17}\}$ | 1 | 220 | C. P. | 80.7 | 65.8 | 51.2 | 58.8 | 67.7 | 78.1 | 48.2 | 67.2 | 13.3 |
| $\{\pi_{605}\}$ | 1 | 1040 | Precal. | 84.7 | 63.2 | 58.5 | 49.5 | 82.3 | 78.1 | 62.5 | 71.8 | 14.6 |
| $\{\pi_{606}\}$ | 1 | 460 | N. T. | 83.9 | 63.2 | 53.7 | 49.5 | 58.1 | 75.6 | 46.4 | 64.4 | 14.2 |
| $\{\pi_{1201}\}$ | 1 | 940 | Geo. | 89.5 | 68.4 | 58.5 | 53.6 | 79.0 | 73.2 | 62.5 | 71.4 | 16.3 |
| $\{\pi_{1207}\}$ | 1 | 100 | Geo. | 67.7 | 50.0 | 43.9 | 41.2 | 53.2 | 63.4 | 42.7 | 54.0 | 9.6 |
| $\{\pi_{1208}\}$ | 1 | 240 | C. P. | 58.1 | 55.3 | 43.9 | 32.0 | 40.3 | 48.8 | 32.1 | 45.0 | 8.8 |
| $\{\pi_{1209}\}$ | 1 | 1140 | Precal. | 86.3 | **71.1** | **65.9** | 55.7 | 75.8 | 76.8 | 64.3 | 72.2 | 17.5 |
| $\{\pi_1 \ldots \pi_{16}\}$ | 16 | 1840 | General | 90.3 | 63.2 | 61.0 | 55.7 | 69.4 | 80.5 | 60.7 | 71.6 | 16.7 |
| $\{\pi_1, \pi_2\}$ | 2 | 1580 | Alg./N.T. | 89.5 | 63.2 | 61.0 | 60.8 | 82.3 | 74.4 | 58.9 | 72.8 | 15.0 |
| $\{\pi_1, \pi_{13}\}$ | 2 | 2000 | Alg./Geo. | **92.7** | **71.1** | 58.5 | 57.7 | 79.0 | **84.2** | **71.4** | **76.0** | 17.9 |

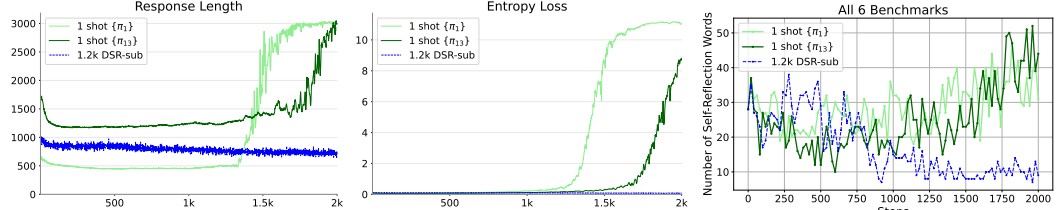

Figure 4: **(Left, Middle) Average response length on training data and entropy loss.** After around 1300/1700 steps, the average response length of 1-shot RLVR with $\pi_1/\pi_{13}$ significantly increases, corresponding to that model tries to solve the single problem with longer CoT reasoning in a more diverse way (Fig. 3, step 1300), which is also confirmed by the increase of entropy loss. These may also indicate the gradual overfitting (Fig. 3, step 1860). **(Right) Number of reflection words detected in evaluation tasks.** The number of reflection words ("rethink", "recheck", and "recalculate") appearing in evaluation tasks increases in 1-shot RLVR with $\pi_1/\pi_{13}$, especially after around 1250 steps, matching the increase of response length. On the other hand, RLVR with DSR-sub contains fewer reflection words as the training progresses.

that their self-reflection process often appears with words *"rethink"*, *"recheck"* and *"recalculate"*. Therefore, we count the number of responses that contain these three words when evaluating 6 mathematical reasoning tasks. The results are in Fig. 4. *First*, after around 1.3k steps, the response length and entropy loss increase significantly, which may imply the attempt of diverse output patterns or overfitting (Fig. 3). *Second*, for the evaluation task, the base model itself already exhibits self-reflection processes, which supports the observation in recent works [13, 21]. *Third*, the number of self-recheck processes increases at the later stages of 1-shot RL training, which again confirms that the model generalizes well on test data and shows more complex reasoning processes even after it

Table 4: **1(few)-shot RLVR is viable for different models and RL algorithm.** "Random" denotes the 16 examples randomly sampled from 1.2k DSR-sub. Format reward (Appendix C.2.3) serves as a baseline for format correction. More details are in Appendix C.1, and we also include the results of Qwen2.5-Math-1.5B-Instruct and Qwen2.5-1.5B in Appendix C.1.2.

| RL Dataset | Dataset Size | MATH 500 | AIME 2024 | AMC 2023 | Minerva Math | Olympiad-Bench | AIME 2025 | Avg. |
|---|---|---|---|---|---|---|---|---|
| **Qwen2.5-Math-7B [24] + GRPO** | | | | | | | | |
| NA | NA | 51.0 | 12.1 | 35.3 | 11.0 | 18.2 | 6.7 | 22.4 |
| DSR-sub | 1209 | 78.6 | 25.8 | 62.5 | 33.8 | 41.6 | **14.6** | **42.8** |
| Format Reward | 1209 | 65.8 | 24.2 | 54.4 | 24.3 | 30.4 | 6.7 | 34.3 |
| $\{\pi_1\}$ | 1 | **79.2** | 23.8 | 60.3 | 27.9 | 39.1 | 10.8 | 40.2 |
| $\{\pi_1, \pi_{13}\}$ | 2 | **79.2** | 21.7 | 58.8 | 35.3 | 40.9 | 12.1 | 41.3 |
| $\{\pi_1, \pi_2, \pi_{13}, \pi_{1209}\}$ | 4 | 78.6 | 22.5 | 61.9 | **36.0** | **43.7** | 12.1 | 42.5 |
| Random | 16 | 76.0 | 22.1 | **63.1** | 31.6 | 35.6 | 12.9 | 40.2 |
| $\{\pi_1, \ldots, \pi_{16}\}$ | 16 | 77.8 | **30.4** | 62.2 | 35.3 | 39.9 | 9.6 | 42.5 |
| **Llama-3.2-3B-Instruct [26] + GRPO** | | | | | | | | |
| NA | NA | 40.8 | 8.3 | 25.3 | 15.8 | 13.2 | 1.7 | 17.5 |
| DSR-sub | 1209 | 43.2 | **11.2** | 27.8 | 19.5 | 16.4 | 0.8 | 19.8 |
| $\{\pi_1\}$ | 1 | 45.8 | 7.9 | 25.3 | 16.5 | 17.0 | **1.2** | 19.0 |
| $\{\pi_1, \pi_{13}\}$ | 2 | **49.4** | 7.1 | **31.6** | 18.4 | **19.1** | 0.4 | **21.0** |
| $\{\pi_1, \pi_2, \pi_{13}, \pi_{1209}\}$ | 4 | 46.4 | 6.2 | 29.1 | **21.0** | 15.1 | **1.2** | 19.8 |
| **Qwen2.5-Math-1.5B [24] + PPO** | | | | | | | | |
| NA | NA | 36.0 | 6.7 | 28.1 | 8.1 | 22.2 | 4.6 | 17.6 |
| DSR-sub | 1209 | 72.8 | 19.2 | 48.1 | 27.9 | 35.0 | 9.6 | 35.4 |
| $\{\pi_1\}$ | 1 | 72.4 | 11.7 | 51.6 | 26.8 | 33.3 | 7.1 | 33.8 |
| **DeepSeek–R1–Distill–Qwen–1.5B [2] + GRPO (Eval=32k)** | | | | | | | | |
| NA | NA | 82.9 | 29.8 | 63.2 | 26.4 | 43.1 | 23.9 | 44.9 |
| DSR-sub | 1209 | 84.5 | 32.7 | 70.1 | 29.5 | 46.9 | 27.8 | 48.6 |
| $\{\pi_1\}$ | 1 | 83.9 | 31.0 | 66.1 | 28.3 | 44.6 | 24.1 | 46.3 |
| $\{\pi_1, \pi_2, \pi_{13}, \pi_{1209}\}$ | 4 | 84.8 | 32.2 | 66.6 | 27.7 | 45.5 | 24.8 | 46.9 |
| $\{\pi_1, \ldots, \pi_{16}\}$ | 16 | 84.5 | 34.3 | 69.0 | 30.0 | 46.9 | 25.2 | 48.3 |

overfits the training data. Interestingly, for the 1.2k DeepScaleR subset, the frequency of reflection slightly decreases as the training progresses, matching the decreasing response length.

## 3.3 1/Few-shot RLVR on Other Models/Algorithms

We further investigate whether 1(few)-shot RLVR is feasible for other models and RL algorithms. We consider setup mentioned in Sec. 3.1, and the results are shown in Tab. 4 (Detailed results on each benchmark are in Appendix C.1). We can see (1) for Qwen2.5-Math-7B, 1-shot RLVR with $\pi_1$ improves average performance by 17.8% (5.9% higher than format-reward baseline), and 4-shot RLVR performs as well as RLVR with DSR-sub. Moreover, $\{\pi_1, \ldots, \pi_{16}\}$ performs better than the subset consisting of 16 randomly sampled examples. (2) For Llama-3.2-3B-Instruct, the absolute gain from RLVR is smaller, but 1(few)-shot RLVR still matches or surpasses (e.g., $\{\pi_1, \pi_{13}\}$) the performance of full-set RLVR. We also show the instability of the RLVR process on Llama-3.2-3B-Instruct in Appendix C.1. (3) RLVR with $\pi_1$ using PPO also works for Qwen2.5-Math-1.5B with PPO. (4) For DeepSeek-R1-Distill-Qwen-1.5B, the performance gap between few-shot and full-set RLVR is larger. Nevertheless, few-shot RLVE still yield improvement. More results are in Appendix C.

## 4 Analysis

Table 5: **Ablation study of loss function and label correctness.** Here we use Qwen2.5-Math-1.5B and example $\pi_1$. "+" means the component is added. "Convergence" denotes if the training accuracy saturates (e.g. Fig. 2). "-0.003" is the coefficient of entropy loss (default -0.001). We report the best model performance on each benchmark separately (Appendix B.3). **(1) Rows 1-8**: The improvement of 1(few)-shot RLVR is mainly attributed to policy gradient loss, and it can be enhanced by adding entropy loss. **(2) Rows 9-10**: Simply adding entropy loss alone can still improve MATH500, but still worse than the format reward baseline (Tab. 3, MATH500: 65.6, AIME24: 10.0). **(3) Rows 5,11-13**: further investigation into how different labels affect test performance.

| Row | Policy Loss | Weight Decay | KL Loss | Entropy Loss | Label | Training Convergence | MATH 500 | AIME 2024 |
|---|---|---|---|---|---|---|---|---|
| 1 | | | | | 12.8 | NO | 39.8 | 7.5 |
| 2 | + | | | | 12.8 | YES | 71.8 | 15.4 |
| 3 | + | + | | | 12.8 | YES | 71.4 | 16.3 |
| 4 | + | + | + | | 12.8 | YES | 70.8 | 15.0 |
| 5 | + | + | + | + | 12.8 | YES | 74.8 | **17.5** |
| 6 | + | + | + | +, −0.003 | 12.8 | YES | 73.6 | 15.4 |
| 7 | + | | | + | 12.8 | YES | **75.6** | 17.1 |
| 8 | | + | + | | 12.8 | NO | 39.0 | 10.0 |
| 9 | | + | + | + | 12.8 | NO | 65.4 | 7.1 |
| 10 | | | | + | 12.8 | NO | 63.4 | 8.8 |
| 11 | + | + | + | + | 12.7 | YES | 73.4 | 17.9 |
| 12 | + | + | + | + | 4 | YES | 57.0 | 9.2 |
| 13 | + | + | + | + | 929725 | NO | 64.4 | 9.6 |

In this section, we concentrate on exploring the potential mechanisms that allow RLVR to work with only one or a few examples. We hope the following analyses can provide some insight for future works. Additional experiments and discussions about the format correction (Appendix C.2.3), prompt modification (Appendix C.2.5) and the reasoning capabilities of base models (Appendix D) are included in supplementary materials.

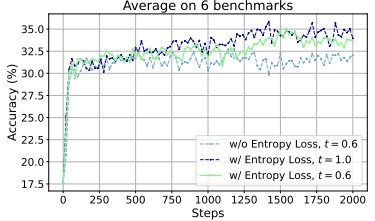

Figure 5: **Encouraging exploration can improve post-saturation generalization.** $t$ is the temperature parameter for training rollouts.

### 4.1 Ablation Study:
### Policy Gradient Loss is the Main Contributor, and Entropy Loss Further Improve Post-Saturation Generalization

As discussed in Sec. 3.2.2, 1-shot RLVR shows the property of post-saturation generalization. This phenomenon is similar to "grokking" [36, 37], which shows that neural networks first memorize/overfit the training data but still perform poorly on the test set, while suddenly improve generalization after many training steps. A natural question is raised: *Is the performance gain from 1-shot RLVR related to the "grokking" phenomenon?* To answer this question, noting "grokking" is strongly affected by regularization [36, 38–41] like weight decay, we conduct an ablation study by removing or changing the components of the loss function one by one to see how each of them contributes to the improvement.

The results are shown in Tab. 5 (Test curves are in Appendix C.2.1). We see that if we only add policy gradient loss (Row 2) with $\pi_1$, we already get results close to that of the full loss training (Row 5). In addition, further adding weight decay (Row 3) and KL divergence loss (Row 4) has no significant impact on model performance, while adding entropy loss (Row 5) can further bring 4.0% improvement for MATH500 and 2.5% for AIME24. Here we need to be careful about the weight of the entropy loss, as a too large coefficient (Row 6) might make the training more unstable. These observations support that **the feasibility of 1(few)-shot RLVR is mainly attributed to policy gradient loss, rather than weight decay, distinguishing it from "grokking"**, which should be significantly affected by weight decay. To double check this, we show that only adding weight decay and KL divergence (Row 8) has little influence on model performance, while using only policy gradient loss and entropy loss (Row 7) behaves almost the same as the full GRPO loss.

Moreover, we also argue that encouraging greater diversity in model outputs—for instance, adding proper entropy loss — can enhance post-saturation generalization in 1-shot RLVR. As shown in Fig. 5, without entropy loss, model performance under 1-shot RLVR shows limited improvement beyond step 150, coinciding with the point at which training accuracy saturates (Fig. 2, Left). By adding entropy loss, the model achieves an average improvement of 2.3%, and further increasing

the temperature to $t = 1.0$ yields an additional 0.8% gain. More discussions about entropy loss and post-saturation generalization are in Appendix C.2.2.

### 4.2 Entropy-Loss-Only Training & Label Correctness

In Tab. 3, we find that when using $\pi_{1207}$ and $\pi_{1208}$, it is difficult for model to output the ground truth label and receive rewards during 1-shot RLVR training, resulting in a very sparse policy gradient signal. Nevertheless, they still outperform the base model, although their performance remains lower than that of the format-reward baseline. To investigate this, we remove the policy loss from the full GRPO loss (Tab. 5, Row 9) or even retain only the entropy loss (Row 10), and again observe similar improvement. Furthermore, this phenomenon also happens on Qwen2.5-Math-7B and Llama-3.2-3B-Instruct, although only improve at the first several steps. These results implies entropy loss may independently contribute to performance gains from format correction, which, although much smaller than those from policy loss, are still nontrivial.

Table 6: **Training with only entropy loss using $\pi_1$ can partially improve base model performance, but still perform worse than format-reward baseline.** Details are in Tab. 13.

| Model | M500 | Avg. |
|---|---|---|
| **Qwen2.5-Math-1.5B** | 36.0 | 17.6 |
| +Entropy Loss, 20 steps | 63.4 | 25.0 |
| Format Reward | 65.0 | 28.7 |
| **Llama-3.2-3B-Instruct** | 40.8 | 17.5 |
| +Entropy Loss, 10 steps | 47.8 | 19.5 |
| **Qwen2.5-Math-7B** | 51.0 | 22.4 |
| +Entropy Loss, 4 steps | 57.2 | 25.0 |
| Format Reward | 65.8 | 34.3 |

Moreover, we conduct an experiment by altering the label to (1) the correct one ("12.7," Row 11), (2) an incorrect one that model can still overfit ("4," Row 12), and (3) an incorrect one that the model can neither guess nor overfit ("9292725," Row 13). We compare them with (4) the original label ("12.8," Row 5). Interestingly, we find the performance rankings are (1) $\approx$ (4) $>$ (3) $>$ (2). This suggests that slight inaccuracies in the label do not significantly impair 1-shot RLVR performance. However, if the incorrect label deviates substantially while remaining guessable and overfittable, the resulting performance can be even worse than using a completely incorrect and unguessable label, which behaves similarly to training with entropy loss alone (Row 10). In Appendix C.2.4, we also discuss label robustness on full-set RLVR by showing that if too many data in the dataset are assigned random wrong labels, full-set RLVR can perform worse than 1-shot RLVR.

## 5 Conclusion

In this work, we show that 1-shot RLVR is sufficient to trigger substantial improvements in reasoning tasks, even matching the performance of RLVR with thousands of examples. The empirical results reveal not only improved task performance but also additional observations such as post-saturation generalization, cross-category generalization, more frequent self-reflection and also additional analysis. These findings suggest that the reasoning capability of the model is already buried in some base models, and encouraging exploration on a very small amount of data is capable of generating useful RL training signals for igniting these LLM's reasoning capability. It also demonstrates the anti-overfitting property of the RLVR algorithm with zero-mean advantage, as we can train on a single example millions of times without performance degradation. Our work also emphasizes the importance of better selection and collection of data for RLVR. We discuss directions for future work in Appendix D.4, and also discuss limitations in Appendix D.1.

## 6 Acknoledgements

We thank Lifan Yuan, Hamish Ivison, Rulin Shao, Shuyue Stella Li, Rui Xin, Scott Geng, Pang Wei Koh, Kaixuan Huang, Mickel Liu, Jacqueline He, Noah Smith, Jiachen T. Wang, Yifang Chen, and Weijia Shi for very constructive discussions. YW and ZZ acknowledge the support of Amazon AI Ph.D. Fellowship. SSD acknowledges the support of NSF IIS-2110170, NSF DMS-2134106, NSF CCF-2212261, NSF IIS-2143493, NSF CCF-2019844, NSF IIS-2229881, and the Sloan Research Fellowship.

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

# Contents

## A Related Work

**Reinforcement Learning with Verifiable Reward (RLVR).** RLVR, where the reward is computed by a rule-based verification function, has been shown to be effective in improving the reasoning capabilities of LLMs. The most common practice of RLVR when applying reinforcement learning to LLMs on mathematical reasoning datasets is to use answer matching: the reward function outputs a binary signal based on if the model's answer matches the gold reference answer [4, 5, 2, 3, 42–44]. This reward design avoids the need for outcome-based or process-based reward models, offering a simple yet effective approach. The success of RLVR is also supported by advancements in RL algorithms, including value function optimization or detail optimization in PPO [7] (e.g., VinePPO [9], VC-PPO [10], VAPO [12]), stabilization and acceleration of GRPO [2] (e.g., DAPO [11], Dr. GRPO [13], GRPO+[14], SRPO [16]), and integration of various components (e.g., REINFORCE++[15]). There are also some recent works that focus on RLVR with minimal human supervision (without using labeled data or even problems), such as Absolute-Zero [45], EMPO [46], and TTRL [47].

**Data Selection for LLM Post-Training.** The problem of data selection for LLM post-training has been extensively studied in prior work [48], with most efforts focusing on data selection for supervised fine-tuning (instruction tuning). These approaches include LLM-based quality assessment [49], leveraging features from model computation [50], gradient-based selection [51], and more. Another line of work [52–54] explores data selection for human preference data in Reinforcement Learning from Human Feedback (RLHF) [55]. Data selection for RLVR remains relatively unexplored. One attempt is LIMR [19], which selects 1.4k examples from an 8.5k full set for RLVR to match performance; however, unlike our work, they do not push the limits of training set size to the extreme case of just a single example. Another closely related concurrent work [56] shows that RLVR using PPO with only 4 examples can already yield very significant improvements; however, they do not systematically explore this observation, nor do they demonstrate that such an extremely small training set can actually match the performance of using the full dataset.

## B Experiment Setup

### B.1 Details of Loss Function

As said in the main paper, we contain three components in the GRPO loss function following verl [22] pipeline: policy gradient loss, KL divergence, and entropy loss. Details are as follows. For each question $q$ sampled from the Question set $P(Q)$, GRPO samples a group of outputs $\{o_1, o_2, \ldots, o_G\}$ from the old policy model $\pi_{\theta_{\text{old}}}$, and then optimizes the policy model $\pi_\theta$ by minimizing the following

loss function:

$$\mathcal{L}_{\text{GRPO}}(\theta) = \mathbb{E}_{\substack{q \sim P(Q) \\ \{o_i\}_{i=1}^G \sim \pi_{\theta_{\text{old}}}(O|q)}} \left[ \mathcal{L}'_{\text{PG-GRPO}}(\cdot, \theta) + \beta \mathcal{L}'_{\text{KL}}(\cdot, \theta, \theta_{\text{ref}}) + \alpha \mathcal{L}'_{\text{Entropy}}(\cdot, \theta) \right], \quad (3)$$

where $\beta$ and $\alpha$ are hyper-parameters (in general $\beta > 0$, $\alpha < 0$), and "$\cdot$" is the abbreviation of sampled prompt-responses: $\{q, \{o_i\}_{i=1}^G\}$. The policy gradient loss and KL divergence loss are:

$$\mathcal{L}'_{\text{PG-GRPO}}(q, \{o_i\}_{i=1}^G, \theta) = -\frac{1}{G} \sum_{i=1}^G \left( \min \left( \frac{\pi_\theta(o_i|q)}{\pi_{\theta_{\text{old}}}(o_i|q)} A_i, \text{clip}\left( \frac{\pi_\theta(o_i|q)}{\pi_{\theta_{\text{old}}}(o_i|q)}, 1 - \varepsilon, 1 + \varepsilon \right) A_i \right) \right) \quad (4)$$

$$\mathcal{L}'_{\text{KL}}(q, \{o_i\}_{i=1}^G, \theta, \theta_{\text{ref}}) = \mathbb{D}_{\text{KL}}(\pi_\theta \| \pi_{\theta_{\text{ref}}}) = \frac{\pi_{\theta_{\text{ref}}}(o_i|q)}{\pi_\theta(o_i|q)} - \log \frac{\pi_{\theta_{\text{ref}}}(o_i|q)}{\pi_\theta(o_i|q)} - 1, \quad (5)$$

Here $\theta_{\text{ref}}$ is the reference model, $\varepsilon$ is a hyper-parameter of clipping threshold. Notably, we use the approximation formulation of KL divergence [57], which is widely used in previous works [8, 2]. Besides, $A_i$ is the group-normalized advantage defined below.

$$A_i = \frac{r_i - \text{mean}(\{r_1, r_2, \dots, r_G\})}{\text{std}(\{r_1, r_2, \dots, r_G\})}. \quad i \in [G] \quad (6)$$

Since we focus on math questions, we let the reward $r_i$ be the 0-1 accuracy score, and $r_i$ is 1 if and only if the response $o_i$ gets the correct answer to the question $q$. What's more, the entropy loss $\mathcal{L}'_{\text{Entropy}}$ calculates the average per-token entropy of the responses, and its coefficient $\alpha < 0$ implies the encouragement of more diverse responses.

The details of entropy loss are as follows. For each query $q$ and set of outputs $\{o_i\}_{i=1}^G$, the model produces logits $X$ that determine the policy distribution $\pi_\theta$. These logits $X$ are the direct computational link between inputs $q$ and outputs $o$ - specifically, the model processes $q$ to generate logits $X$, which after softmax normalization give the probabilities used to sample each token in the outputs $o$. The entropy loss is formally defined below.

$$\mathcal{L}'_{\text{Entropy}}(q, \{o_i\}_{i=1}^G, \theta) = \frac{\sum_{b,s} M_{b,s} \cdot H_{b,s}(X)}{\sum_{b,s} M_{b,s}} \quad (7)$$

Here $M_{b,s}$ represents the response mask indicating which tokens contribute to the loss calculation (excluding padding and irrelevant tokens), with $b$ indexing the batch dimension and $s$ indexing the sequence position. The entropy $H_{b,s}(X)$ is computed from the model's logits $X$:

$$H_{b,s}(X) = \log\left(\sum_v e^{X_{b,s,v}}\right) - \sum_v p_{b,s,v} \cdot X_{b,s,v} \quad (8)$$

where $v$ indexes over the vocabulary tokens (i.e., the possible output tokens from the model's vocabulary), and the probability distribution is given by $p_{b,s,v} = \text{softmax}(X_{b,s})_v = \frac{e^{X_{b,s,v}}}{\sum_{v'} e^{X_{b,s,v'}}}$.

## B.2 Training Dataset

**DeepScaleR-sub.** DeepScaleR-Preview- Dataset [18] consists of approximately 40,000 unique mathematics problem-answer pairs from AIME (1984-2023), AMC (pre-2023), and other sources including Omni-MATH [58] and Still [59]. The data processing pipeline includes extracting answers using Gemini-1.5-Pro-002, removing duplicate problems through RAG with Sentence-Transformers embeddings, and filtering out questions that cannot be evaluated using SymPy to maintain a clean training set. We randomly select a subset that contains 1,209 examples referred to as "DSR-sub".

**MATH.** Introduced in [27], this dataset contains 12,500 challenging competition mathematics problems designed to measure advanced problem-solving capabilities in machine learning models. Unlike standard mathematical collections, MATH features complex problems from high school mathematics competitions spanning subjects including Prealgebra, Algebra, Number Theory, Counting and Probability, Geometry, Intermediate Algebra, and Precalculus, with each problem assigned a difficulty level from 1 to 5 and accompanied by detailed step-by-step solutions. It's partitioned into a training subset comprising 7,500 problems (60%) and a test subset containing 5,000 problems (40%).

## B.3 Evaulation Dataset

All evaluation sets are drawn from the Qwen2.5-Math evaluation repository[6], with the exception of AIME2025[7]. We summarize their details as follows:

**MATH500.** MATH500, developed by OpenAI [29], comprises a carefully curated selection of 500 problems extracted exclusively from the test partition (n=5,000) of the MATH benchmark [27]. It is smaller, more focused, and designed for efficient evaluation.

**AIME 2024/2025.** The AIME 2024 and 2025 datasets are specialized benchmark collections, each consisting of 30 problems from the 2024 and 2025 American Invitational Mathematics Examination (AIME) I and II, respectively [30].

**AMC 2023.** AMC 2023 dataset consists of 40 problems, selected from two challenging mathematics competitions (AMC 12A and 12B) for students grades 12 and under across the United States [31]. These AMC 12 evaluates problem-solving abilities in secondary school mathematics, covering topics such as arithmetic, algebra, combinatorics, geometry, number theory, and probability, with all problems solvable without calculus.

**Minerva Math.** Implicitly introduced in the paper "Solving Quantitative Reasoning Problems with Language Models" [32] as OCWCourses, Minerva Math consists of 272 undergraduate-level STEM problems harvested from MIT's OpenCourseWare, specifically designed to evaluate multi-step scientific reasoning capabilities in language models. Problems were carefully curated from courses including solid-state chemistry, information and entropy, differential equations, and special relativity, with each problem modified to be self-contained with clearly-delineated answers that are automatically verifiable through either numeric (191 problems) or symbolic solutions (81 problems).

**OlympiadBench.** OlympiadBench [33]is a large-scale, bilingual, and multimodal benchmark designed to evaluate advanced mathematical and physical reasoning in AI systems. It contains 8,476 Olympiad-level problems, sourced from competitions and national exams, with expert-annotated step-by-step solutions. The subset we use for evaluation consists of 675 open-ended text-only math competition problems in English.

We also consider other non-mathematical reasoning tasks: ARC-Challenge and ARC-Easy [34].

**ARC-Challenge/Easy.** The ARC-Challenge benchmark represents a subset of 2,590 demanding science examination questions drawn from the broader ARC (AI2 Reasoning Challenge) [34] collection, specifically selected because traditional information retrieval and word co-occurrence methods fail to solve them correctly. This challenging evaluation benchmark features exclusively text-based, English-language multiple-choice questions (typically with four possible answers) spanning diverse grade levels, designed to assess science reasoning capabilities rather than simple pattern matching or information retrieval. The complementary ARC-Easy [34] subset contains 5197 questions solvable through simpler approaches. We use 1.17k test split for ARC-Challenge evaluation and 2.38k test split for ARC-Easy evaluation, respectively.

## B.4 More Training Details

For DeepSeek-R1-Distill-Qwen-1.5B, we let the maximum response length be 8192, following the setup of stage 1 in DeepScaleR [18]. The learning rate is set to 1e-6. The coefficient of weight decay is set to 0.01 by default. We store the model checkpoint every 20 steps for evaluation, and use 8 A100 GPUs for each experiment. For Qwen2.5-Math-1.5B, Qwen2.5-Math-7B, Llama-3.2-3B-Instruct, and DeepSeek-R1-Distill-Qwen-1.5B, we train for 2000, 1000, 1000, and 1200 steps, respectively, unless the model has already shown a significant drop in performance. We use the same approach as DeepScaleR [18] (whose repository is also derived from the verl) to save the model in safetensor format to facilitate evaluation.

## B.5 More Evaluation Details

In evaluation, the maximum number of generated tokens is set to be 3072 by default. For Qwen-based models, we use the "`qwen25-math-cot`" prompt template in evaluation. For Llama and

---

[6]https://github.com/QwenLM/Qwen2.5-Math
[7]https://huggingface.co/datasets/opencompass/AIME2025

Table 7: **Difference between model downloaded from Hugging Face and initial checkpoint saved by verl/deepscaler pipeline.** Since the performance of stored initial checkpoint has some randomness, we still use the original downloaded model for recording initial performance.

| Model | MATH 500 | AIME24 2024 | AMC23 2023 | Minerva Math | Olympiad-Bench | AIME 2025 | Avg. |
|---|---|---|---|---|---|---|---|
| **Qwen2.5-Math-1.5B [24]** | | | | | | | |
| Hugging Face Model | 36.0 | 6.7 | 28.1 | 8.1 | 22.2 | 4.6 | 17.6 |
| Stored Initial Checkpoint | 39.6 | 8.8 | 34.7 | 8.5 | 22.7 | 3.3 | 19.6 |
| **Qwen2.5-Math-7B [24]** | | | | | | | |
| Hugging Face Model | 51.0 | 12.1 | 35.3 | 11.0 | 18.2 | 6.7 | 22.4 |
| Stored Initial Checkpoint | 52.0 | 14.6 | 36.6 | 12.1 | 18.1 | 4.2 | 22.9 |
| **Llama-3.2-3B-Instruct [26]** | | | | | | | |
| Hugging Face Model | 40.8 | 8.3 | 25.3 | 15.8 | 13.2 | 1.7 | 17.5 |
| Stored Initial Checkpoint | 41.0 | 7.1 | 28.4 | 16.9 | 13.0 | 0.0 | 17.7 |

distilled models, we use their original chat templates. We set the evaluation seed to 0 and `top_p` to 1 by default. For evaluation on DeepSeek-R1-Distill-Qwen-1.5B, following DeepSeek-R1 [2] and DeepScaleR [18], we set the temperature to 0.6 and `top_p` to 0.95, and use `avg@16` for MATH500, Minerva Math, and OlympiadBench, and `avg@64` for AIME24, AIME25, and AMC23. Since our training length is 8192, we provide results for both 8192 (8k) and 32768 (32k) evaluation lengths (Appendix C.1.6). By default, we report the performance of the checkpoint that obtains the best average performance on 6 benchmarks. But in Sec. 3.2.3 and Sec. 4.1, since we only evaluate MATH500 and AIME2024, we report the best model performance on each benchmark separately, i.e., the best MATH500 checkpoint and best AIME2024 checkpoint can be different (This will not influence our results, as in Tab. 9 and Tab. 11, we still obtain similar conclusions as in main paper.) We use 4 GPUs for the evaluation. Finally we mention that there are slightly performance difference on initial model caused by numerical precision, but it does not influence our conclusions (Appendix B.6).

## B.6   Performance Difference on Initial Model

We mention that there is a precision inconsistency between models downloaded from Hugging Face repositories and initial checkpoints saved by the verl/deepscaler reinforcement learning pipeline in Tab. 7. This discrepancy arises from the verl/DeepScaleR pipeline saving checkpoints with float32 precision, whereas the original base models from Hugging Face utilize bfloat16 precision.

The root cause appears to be in the model initialization process within the verl framework. The fsdp_workers.py [8] file in the verl codebase reveals that models are deliberately created in float32 precision during initialization, as noted in the code comment: "note that we have to create model in fp32. Otherwise, the optimizer is in bf16, which is incorrect". This design choice was likely made to ensure optimizer stability during training. When examining the checkpoint saving process, the precision setting from initialization appears to be preserved, resulting in saved checkpoints retaining float32 precision rather than the original bfloat16 precision of the base model.

Our empirical investigation demonstrates that modifying the `torch_dtype` parameter in the saved `config.json` file to match the base model's precision (specifically, changing from `float32` to `bfloat16`) successfully resolves the observed numerical inconsistency. Related issues are documented in the community[9], and we adopt the default settings of the verl pipeline in our experiments.

Table 8: **Detailed 1/2-shot RLVR performance for Qwen2.5-Math-1.5B.** Results are reported for the checkpoint achieving the best average across 6 math benchmarks (Fig. 1). Models' best individual benchmark results are listed in Tab. 9. Format reward (Appendix C.2.3) serves as a baseline for format correction.

| RL Dataset/ Method | Dataset Size | MATH 500 | AIME 2024 | AMC 2023 | Minerva Math | Olympiad-Bench | AIME 2025 | Avg. |
|---|---|---|---|---|---|---|---|---|
| NA | NA | 36.0 | 6.7 | 28.1 | 8.1 | 22.2 | 4.6 | 17.6 |
| MATH | 7500 | 74.4 | **20.0** | **54.1** | 29.0 | 34.1 | 8.3 | **36.7** |
| DSR-sub | 1209 | 73.6 | 17.1 | 50.6 | 32.4 | 33.6 | 8.3 | 35.9 |
| Format Reward | 1209 | 65.0 | 8.3 | 45.9 | 17.6 | 29.9 | 5.4 | 28.7 |
| $\{\pi_1\}$ | 1 | 72.8 | 15.4 | 51.6 | 29.8 | 33.5 | 7.1 | 35.0 |
| $\{\pi_{13}\}$ | 1 | 73.6 | 16.7 | 53.8 | 23.5 | 35.7 | **10.8** | 35.7 |
| $\{\pi_1, \pi_{13}\}$ | 2 | 74.8 | 17.5 | 53.1 | **29.4** | **36.7** | 7.9 | 36.6 |

Table 9: **Detailed 1/2/4-shot RLVR performance for Qwen2.5-Math-1.5B.** Here we record model's best performance on each benchmark independently. "Best Avg. Step" denotes the checkpoint step that model achieves the best average performance (Tab. 8).

| RL Dataset | Dataset Size | MATH 500 | AIME 2024 | AMC 2023 | Minerva Math | Olympiad-Bench | AIME 2025 | Avg. | Best Avg. Step |
|---|---|---|---|---|---|---|---|---|---|
| NA | NA | 36.0 | 6.7 | 28.1 | 8.1 | 22.2 | 4.6 | 17.6 | 0 |
| MATH | 7500 | 75.4 | **20.4** | 54.7 | 29.8 | **37.3** | 10.8 | **36.7** | 2000 |
| DSR-sub | 1209 | 75.2 | 18.8 | 52.5 | **34.9** | 35.1 | **11.3** | 35.9 | 1560 |
| $\{\pi_1\}$ | 1 | 74.0 | 16.7 | 54.4 | 30.2 | 35.3 | 9.2 | 35.0 | 1540 |
| $\{\pi_2\}$ | 1 | 70.6 | 17.1 | 52.8 | 28.7 | 34.2 | 7.9 | 33.5 | 320 |
| $\{\pi_{13}\}$ | 1 | 74.4 | 17.1 | 53.8 | 25.4 | 36.7 | 10.8 | 35.7 | 2000 |
| $\{\pi_{1201}\}$ | 1 | 71.4 | 16.3 | 54.4 | 25.4 | 36.2 | 10.0 | 33.7 | 1120 |
| $\{\pi_{1209}\}$ | 1 | 72.2 | 17.5 | 50.9 | 27.6 | 34.2 | 8.8 | 33.5 | 1220 |
| $\{\pi_1, \pi_{13}\}$ | 2 | **76.0** | 17.9 | 54.1 | 30.9 | 37.2 | 10.8 | 36.6 | 1980 |
| $\{\pi_1, \pi_2, \pi_{13}, \pi_{1209}\}$ | 4 | 74.4 | 16.3 | **56.3** | 32.4 | 37.0 | **11.3** | 36.0 | 1880 |

Table 10: **Results of more models (base and instruct versions) and more training examples (on Qwen2.5-Math-7B).** We record results from checkpoints achieving best average performance. Test curves are in Fig. 10 and Fig. 11. Analysis is in Appendix C.1.2. We can see that on Qwen2.5-Math-7B, different examples have different performance for 1-shot RLVR.

| RL Dataset | Dataset Size | MATH 500 | AIME 2024 | AMC 2023 | Minerva Math | Olympiad-Bench | AIME 2025 | Avg. |
|---|---|---|---|---|---|---|---|---|
| **Qwen2.5-1.5B [24]** | | | | | | | | |
| NA | NA | 3.2 | 0.4 | 3.1 | 2.6 | 1.2 | 1.7 | 2.0 |
| DSR-sub | 1209 | 57.2 | 5.0 | 30.3 | 17.6 | 21.2 | 0.8 | 22.0 |
| $\{\pi_1\}$ | 1 | 43.6 | 0.8 | 14.4 | 12.9 | 17.6 | 0.4 | 15.0 |
| $\{\pi_1, \pi_2, \pi_{13}, \pi_{1209}\}$ | 4 | 46.4 | 2.9 | 15.9 | 14.0 | 19.0 | 0.8 | 16.5 |
| $\{\pi_1, \ldots, \pi_{16}\}$ | 16 | 53.0 | 3.8 | 30.3 | 19.1 | 19.6 | 0.0 | 21.0 |
| **Qwen2.5-Math-1.5B-Instruct [25]** | | | | | | | | |
| NA | NA | 73.4 | 10.8 | 55.0 | 29.0 | 38.5 | 6.7 | 35.6 |
| DSR-sub | 1209 | 75.6 | 13.3 | 57.2 | 31.2 | 39.6 | 12.1 | 38.2 |
| $\{\pi_1\}$ | 1 | 74.6 | 12.1 | 55.3 | 30.9 | 37.9 | 12.1 | 37.1 |
| **Qwen2.5-Math-7B [25]** | | | | | | | | |
| NA | NA | 51.0 | 12.1 | 35.3 | 11.0 | 18.2 | 6.7 | 22.4 |
| DSR-sub | 1209 | 78.6 | 25.8 | **62.5** | 33.8 | **41.6** | **14.6** | **42.8** |
| $\{\pi_1\}$ | 1 | **79.2** | 23.8 | 60.3 | 27.9 | 39.1 | 10.8 | 40.2 |
| $\{\pi_{605}\}$ | 1 | 77.4 | 20.4 | 59.4 | 23.9 | 39.0 | 10.8 | 38.5 |
| $\{\pi_{1209}\}$ | 1 | 76.4 | 16.2 | 55.0 | 30.9 | 41.0 | 5.4 | 37.5 |
| $\{\pi_1, \ldots, \pi_{16}\}$ | 16 | 77.8 | **30.4** | 62.2 | **35.3** | 39.9 | 9.6 | 42.5 |

Table 11: **1(few)-shot RL still works well for different model with different scales.** Here we record model's best performance on each benchmark independently.

| RL Dataset | Dataset Size | MATH 500 | AIME 2024 | AMC 2023 | Minerva Math | Olympiad-Bench | AIME 2025 | Avg. |
|---|---|---|---|---|---|---|---|---|
| **Qwen2.5-Math-7B [24] + GRPO** | | | | | | | | |
| NA | NA | 51.0 | 12.1 | 35.3 | 11.0 | 18.2 | 6.7 | 22.4 |
| DSR-sub | 1209 | 81.0 | 34.6 | 64.6 | 39.7 | 42.2 | 14.6 | 42.8 |
| $\{\pi_1\}$ | 1 | 79.4 | 27.1 | 61.9 | 32.7 | 40.3 | 11.7 | 40.2 |
| $\{\pi_1, \pi_{13}\}$ | 1 | 81.2 | 23.3 | 64.1 | 36.0 | 42.2 | 12.1 | 41.3 |
| $\{\pi_1, \pi_2, \pi_{13}, \pi_{1209}\}$ | 4 | 80.0 | 26.2 | 64.4 | 37.9 | 43.7 | 14.6 | 42.5 |
| Random | 16 | 78.0 | 24.6 | 63.1 | 36.8 | 38.7 | 14.2 | 40.2 |
| $\{\pi_1, \ldots, \pi_{16}\}$ | 16 | 79.2 | 30.4 | 62.2 | 37.9 | 42.4 | 11.7 | 42.5 |
| **Llama-3.2-3B-Instruct [26] + GRPO** | | | | | | | | |
| NA | NA | 40.8 | 8.3 | 25.3 | 15.8 | 13.2 | 1.7 | 17.5 |
| DSR-sub | 1209 | 45.4 | 11.7 | 30.9 | 21.7 | 16.6 | 11.7 | 19.8 |
| $\{\pi_1\}$ | 1 | 46.4 | 8.3 | 27.5 | 19.5 | 18.2 | 1.7 | 19.0 |
| $\{\pi_1, \pi_{13}\}$ | 2 | 49.4 | 9.2 | 31.6 | 20.6 | 20.0 | 2.1 | 21.0 |
| $\{\pi_1, \pi_2, \pi_{13}, \pi_{1209}\}$ | 4 | 48.4 | 9.2 | 29.4 | 23.5 | 17.6 | 1.7 | 19.8 |
| **Qwen2.5-Math-1.5B [24] + PPO** | | | | | | | | |
| NA | NA | 36.0 | 6.7 | 28.1 | 8.1 | 22.2 | 4.6 | 17.6 |
| DSR-sub | 1209 | 73.8 | 21.2 | 52.8 | 32.4 | 36.3 | 10.4 | 35.4 |
| $\{\pi_1\}$ | 1 | 74.0 | 16.7 | 53.8 | 28.3 | 34.1 | 9.2 | 33.8 |

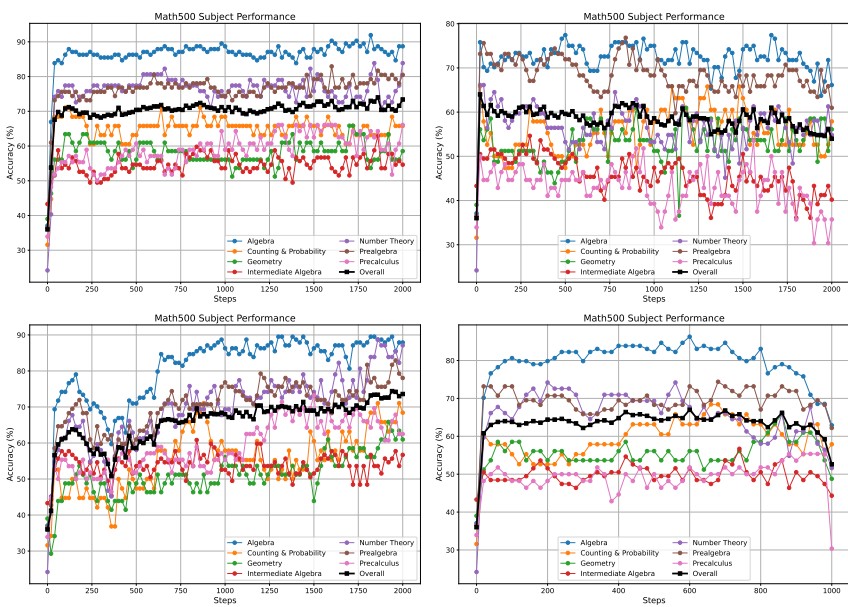

Figure 6: **Different data have large difference on improving MATH500 accuracy, but they all improve various tasks rather than their own task.** From left to right correspond to 1-shot RL on $\pi_1$, $\pi_{11}$, $\pi_{13}$, or $\pi_{16}$. Details are in Tab. 3.

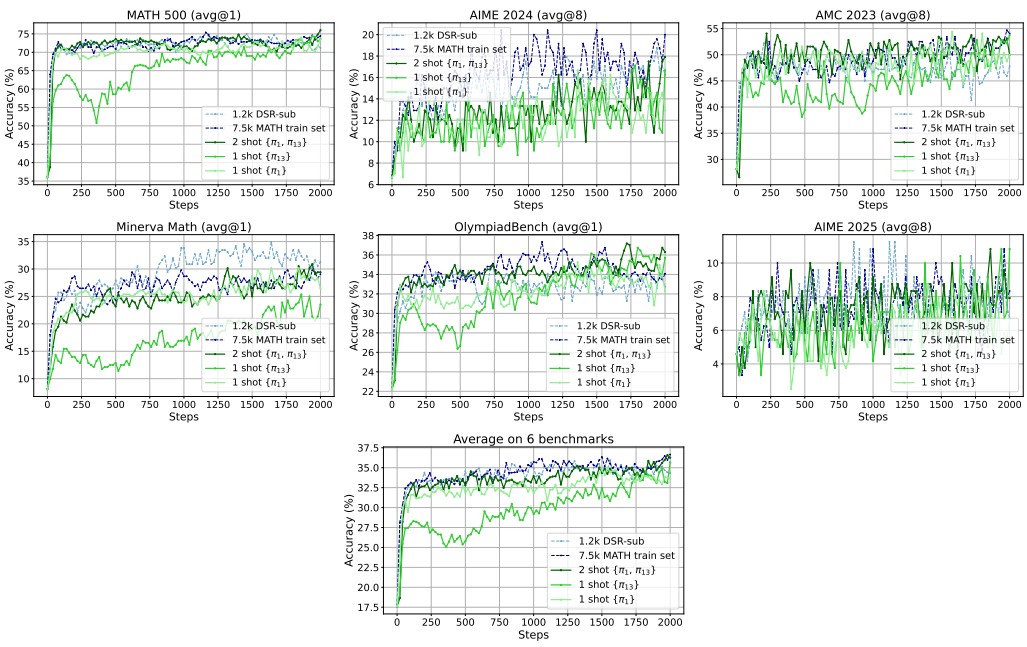

Figure 7: **Detailed results for RLVR on Qwen2.5-Math-1.5B.**

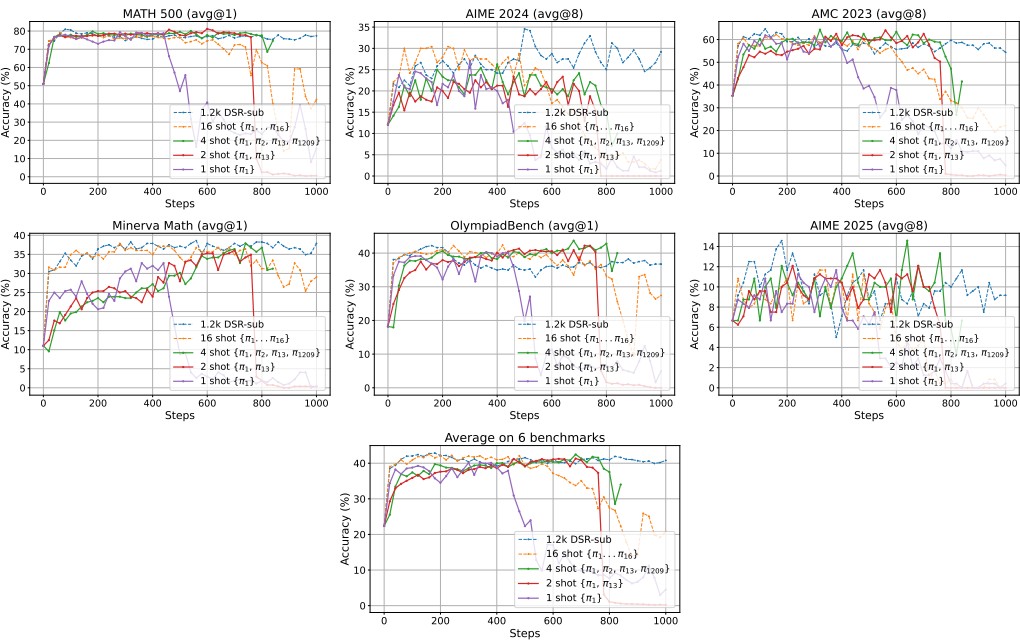

Figure 8: **Detailed results for RLVR on Qwen2.5-Math-7B.**

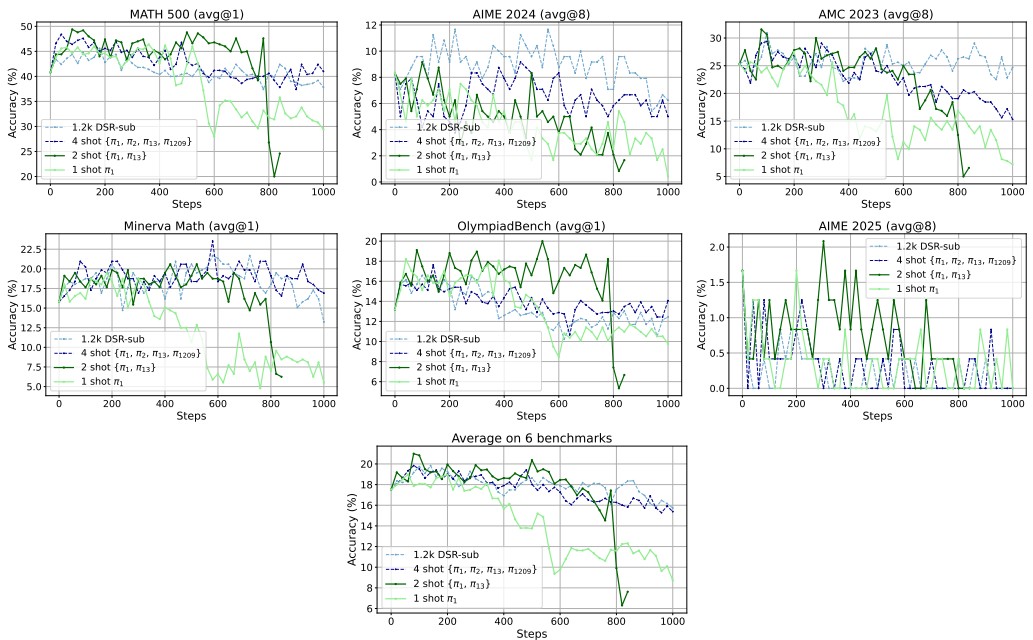

Figure 9: **Detailed results for RLVR on Llama-3.2-3B-Instruct.**

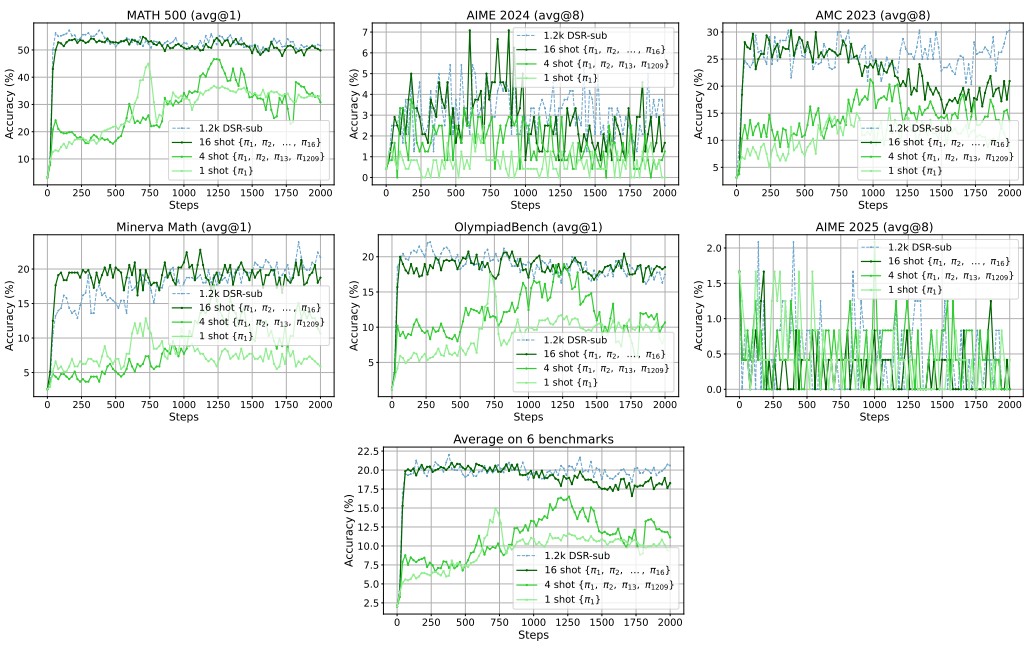

Figure 10: **Detailed results for RLVR on Qwen2.5-1.5B.** The gap between 1-shot RLVR and full-set RLVR is larger, but the 1-shot RLVR still improves a lot from initial model and 16-shot RLVR behaves close to full-set RLVR.

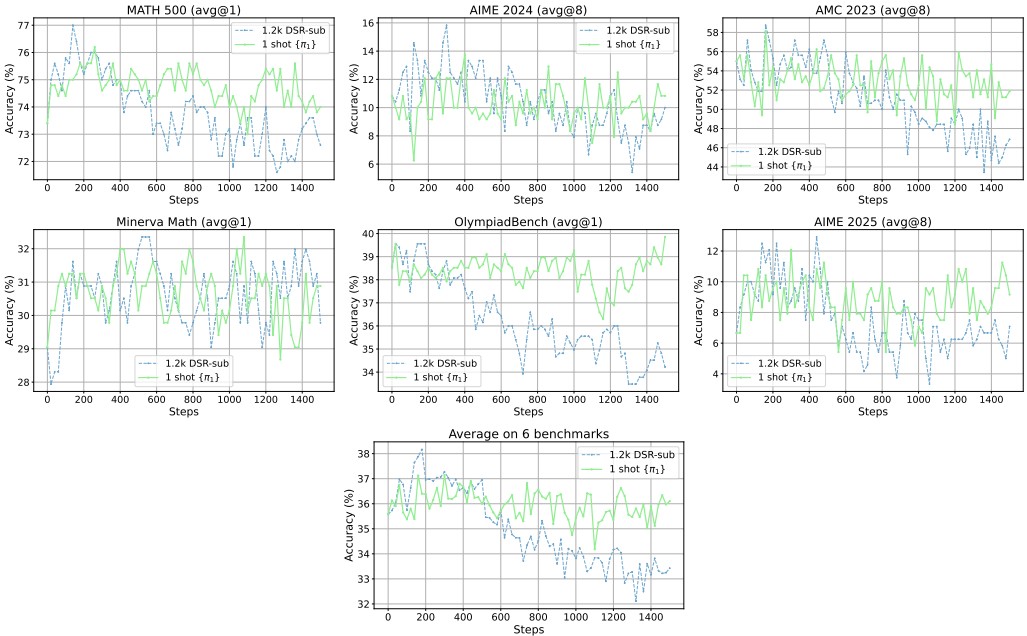

Figure 11: **Detailed results for RLVR on Qwen2.5-Math-1.5B-Instruct.** Interestingly, 1-shot RLVR is more stable than full-set RLVR here.

## C   Evaluation Result

### C.1   Main Experiments

#### C.1.1   Detailed performance on Qwen2.5-Math-1.5B.

In Tab. 8, we show the detailed performance that shown in Fig. 1. Results are reported for the checkpoint achieving the best average performance.

#### C.1.2   Detailed Performance on More Models and Training Examples.

In Tab. 10, we also show the 1(few)-shot RLVR results on the base model (Qwen2.5-1.5B [24]) and instruction model (Qwen2.5-Math-1.5B-Instruct [25]). More detailed test curves are shown in Fig. 10 and Fig. 11. We can see that (1) for Qwen2.5-1.5B, the gap between 1-shot RLVR with $\pi_1$ and full-set RLVR is larger, but the former still improves model performance significantly (e.g., MATH500: 3.2% to 43.6%), and 16-shot RLVR works very closely to full-set RLVR. (2) for Qwen2.5-Math-1.5B-Instruct, both full-set RLVR and 1-shot RLVR have limited improvement as the initial model already has good performance. Interestingly, as shown in Fig. 11, we observe that 1-shot RLVR is more stable than full-set RLVR.

Besides, we also consider other single training examples like $\pi_{605}$ and $\pi_{1209}$ on Qwen2.5-Math-7B. We can see that they behave relatively worse than $\pi_1$, and 16-shot RLVR provides a more consistent approach to closing the performance gap relative to full-set RLVR.

#### C.1.3   Detailed performance with best per-benchmark results

In Tab. 9, we present the detailed 1(few)-shot RLVR results for Qwen2.5-Math-1.5B. Here, we record the model's best performance on each benchmark individually, so their average can be higher than the best overall average performance ("Avg."). We include these results to estimate the upper limit of what the model can achieve on each benchmark. Additionally, we include several examples that, while not performing as well as $\pi_1$ or $\pi_{13}$, still demonstrate significant improvements, such as $\pi_2$, $\pi_{1201}$, and $\pi_{1209}$. We observe that, in general, better results correspond to a larger checkpoint step for best average performance, which may correspond to a longer post-saturation generalization

---

[8]https://github.com/volcengine/verl/blob/main/verl/workers/fsdp_workers.py

[9]https://github.com/volcengine/verl/issues/296

process. Similarly, in Tab. 11, we also include the best per-benchmark results for Qwen2.5-Math-7B, Llama-3.2-3B-Instruct, respectively, together with Qwen2.5-Math-1.5B with PPO training.

### C.1.4 Detailed Test curves on MATH500 for 1-shot RLVR on Qwen2.5-Math-1.5B.

We plot the performance curves for each subject in MATH500 under 1-shot RLVR using different mathematical examples. As shown in Fig. 6, the choice of example leads to markedly different improvements and training dynamics in 1-shot RLVR, highlighting the critical importance of data selection for future few-shot RLVR methods.

### C.1.5 Detailed RLVR results on eacn benchmark over training process.

To better visualize the training process of RLVR and compare few-shot RLVR with full-set RLVR, we show the performance curves for each benchamrk on each model in Fig. 7, 8, 9. It will be interesting to see that if applying 1(few)-shot RLVR for more stable GRPO variants [13, 11, 12, 16] can alleviate this phenomenon. In addition to the conclusions discussed in Sec. 3.3, we also note that Llama3.2-3B-Instruct is more unstable during training, as almost all setups start having performance degradation before 200 steps.

In Appendix C.1.2, we also test the base model and instruction version models in Qwen family. Their test curves are also shown in Fig. 10 and Fig. 11.

### C.1.6 More Evaluation on DeepSeek-R1-Distill-Qwen-1.5B

In Tab. 12 we show the DeepSeek-R1-Distill-Qwen-1.5B results at 8k and 32k evaluation lengths. The experimental setup is illustrated in Appendix B.3.

Table 12: **DeepSeek-R1-Distill-Qwen-1.5B results at 8k and 32k evaluation lengths.** Setup details are in Appendix B.3. "8k→16k→24k" denotes the length extension process in DeepScaleR training.

| RL Dataset | Train Length | MATH 500 | AIME 2024 | AMC 2023 | Minerva Math | Olympiad- Bench | AIME 2025 | Avg. |
|---|---|---|---|---|---|---|---|---|
| colspan | | Eval Length = 8k | | | | | | |
| NA | NA | 76.7 | 20.8 | 51.3 | 23.3 | 35.4 | 19.7 | 37.9 |
| DSR-sub | 8k | 84.4 | 30.2 | **68.3** | 29.2 | 45.8 | 26.7 | 47.4 |
| DeepScaleR (40k DSR) | 8k→16k→24k | **86.3** | **35.2** | 68.1 | **29.6** | **46.7** | **28.3** | **49.0** |
| $\{\pi_1\}$ | 8k | 80.5 | 25.1 | 58.9 | 27.2 | 40.2 | 21.7 | 42.3 |
| $\{\pi_1, \pi_2, \pi_{13}, \pi_{1209}\}$ | 8k | 81.2 | 25.8 | 60.1 | 26.8 | 40.4 | 22.0 | 42.7 |
| $\{\pi_1, \ldots, \pi_{16}\}$ | 8k | 83.3 | 29.6 | 64.8 | 29.3 | 43.3 | 22.8 | 45.5 |
| colspan | | Eval Length = 32k | | | | | | |
| NA | NA | 82.9 | 29.8 | 63.2 | 26.4 | 43.1 | 23.9 | 44.9 |
| DSR-sub | 8k | 84.5 | 32.7 | 70.1 | 29.5 | 46.9 | 27.8 | 48.6 |
| DeepScaleR(40k DSR) | 8k→16k→24k | **87.6** | **41.4** | **73.2** | **30.6** | **49.6** | **31.3** | **52.3** |
| $\{\pi_1\}$ | 8k | 83.9 | 31.0 | 66.1 | 28.3 | 44.6 | 24.1 | 46.3 |
| $\{\pi_1, \pi_2, \pi_{13}, \pi_{1209}\}$ | 8k | 84.8 | 32.2 | 66.6 | 27.7 | 45.5 | 24.8 | 46.9 |
| $\{\pi_1, \ldots, \pi_{16}\}$ | 8k | 84.5 | 34.3 | 69.0 | 30.0 | 46.9 | 25.2 | 48.3 |

## C.2 Analysis

### C.2.1 Test Curves for Ablation Study

In Fig. 12, we can see the test curves for ablation study (Sec. 4.1). We can see that policy gradient loss is the main contributor of 1-shot RLVR. More discussions about format fixing are in Appendix C.2.3.

### C.2.2 Entropy loss

**Detailed results of entropy-loss-only training.** As in Sec. 4.2, we show the full results of entropy-loss-only training in Tab. 13. Training with only entropy loss for a few steps can improve model performance on all math benchmarks except AIME2025. The test curves are in Fig. 12. Notice that the improvement of entropy-loss-only training on Qwen2.5-Math-1.5B is similar to that of RLVR with

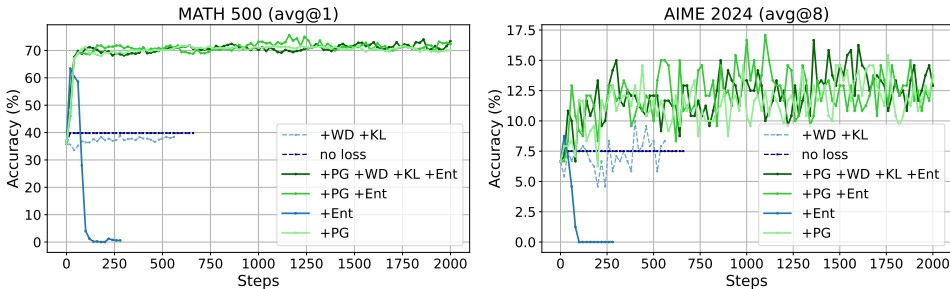

Figure 12: **Test curves for ablation study.** Here we consider adding policy gradient loss (PG), weight decay (WD), KL divergence loss (KL) and entropy loss (Ent) one by one for 1-shot RLVR training on Qwen2.5-Math-1.5B (Sec. 4.1). Especially for only-entropy training, the test performance quickly achieves 0 since too large entropy will result in random output, but before that, the model gets significant improvement from the first several steps, which is close to the results of format-reward RLVR training (Appendix C.2.3). More discussions are in Appendix C.2.3.

Table 13: **Entropy loss alone with $\pi_1$ can improve model performance, but it still underperforms compared to the format-reward baseline (Appendix C.2.3)**.

| Model | MATH 500 | AIME24 2024 | AMC23 2023 | Minerva Math | Olympiad-Bench | AIME 2025 | Avg. |
|---|---|---|---|---|---|---|---|
| **Qwen2.5-Math-1.5B** | 36.0 | 6.7 | 28.1 | 8.1 | 22.2 | 4.6 | 17.6 |
| +Entropy Loss, Train 20 steps | 63.4 | 8.8 | 33.8 | 14.3 | 26.5 | 3.3 | 25.0 |
| Format Reward | 65.0 | 8.3 | 45.9 | 17.6 | 29.9 | 5.4 | 28.7 |
| **Llama-3.2-3B-Instruct** | 40.8 | 8.3 | 25.3 | 15.8 | 13.2 | 1.7 | 17.5 |
| +Entropy Loss, Train 10 steps | 47.8 | 8.8 | 26.9 | 18.0 | 15.1 | 0.4 | 19.5 |
| **Qwen2.5-Math-7B** | 51.0 | 12.1 | 35.3 | 11.0 | 18.2 | 6.7 | 22.4 |
| +Entropy Loss, Train 4 steps | 57.2 | 13.3 | 39.7 | 14.3 | 21.5 | 3.8 | 25.0 |
| Format Reward | 65.8 | 24.2 | 54.4 | 24.3 | 30.4 | 6.7 | 34.3 |

format reward (Appendix C.2.3, Tab. 14), thus we doubt that the effectiveness of entropy-loss-only training may come from format fixing, and we leave the rigorous analysis of this phenomenon for future works.

**Discussion of entropy loss and its function in 1-shot RLVR.** Notably, we observe that the benefit of adding entropy loss for 1-shot RLVR is consistent with conclusions from previous work [60] on the full RLVR dataset, which shows that appropriate entropy regularization can enhance generalization, although it remains sensitive to the choice of coefficient. We conjecture the success of 1-shot RLVR is that the policy gradient loss on the learned example (e.g., $\pi(1)$) actually acts as an implicit regularization by ensuring the correctness of learned training examples when the model tries to explore more diverse responses or strategies, as shown in Fig. 3 (Step 1300). And because of this, both policy loss and entropy loss can contribute to the improvement of 1-shot RLVR. We leave the rigorous analysis to future works.

### C.2.3 (Only) Format Correction?

As discussed in Dr. GRPO [13], changing the template of Qwen2.5-Math models can significantly affect their math performance. In this section, we investigate some critical problems: is (1-shot) RLVR doing format fixing? And if the answer is true, is this the only thing 1-shot RLVR does?

To investigate it, we consider three methods:

**(a). Applying format reward in RLVR.** We first try to apply only format reward for RLVR (i.e., if the verifier can parse the final answer from model output, then it gets 1 reward no matter if the answer is correct or not, otherwise it gets 0 reward), considering both 1-shot and full-set. The results are shown in Tab. 14, and the test curves are shown in Fig. 14 and Fig. 13, respectively.

Notably, we can find that (1) Applying format reward to full-set RLVR and 1-shot RLVR behave very similarly. (2) **applying only format reward is already capable of improving model performance**

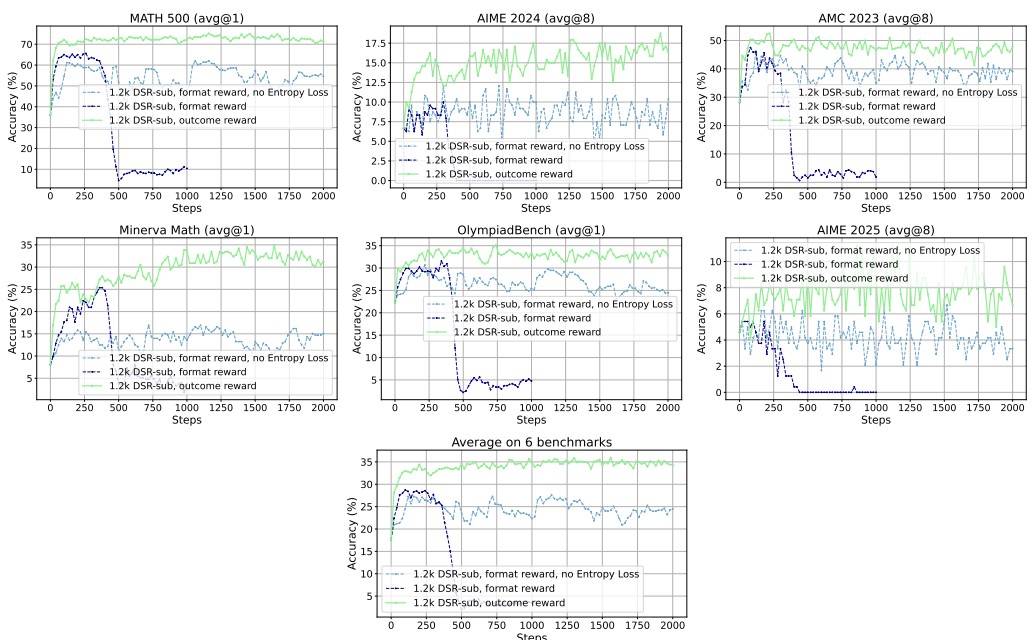

Figure 13: **Comparison between outcome reward and format reward for full-set RLVR with 1.2k DSR-sub on Qwen2.5-Math-1.5B.**

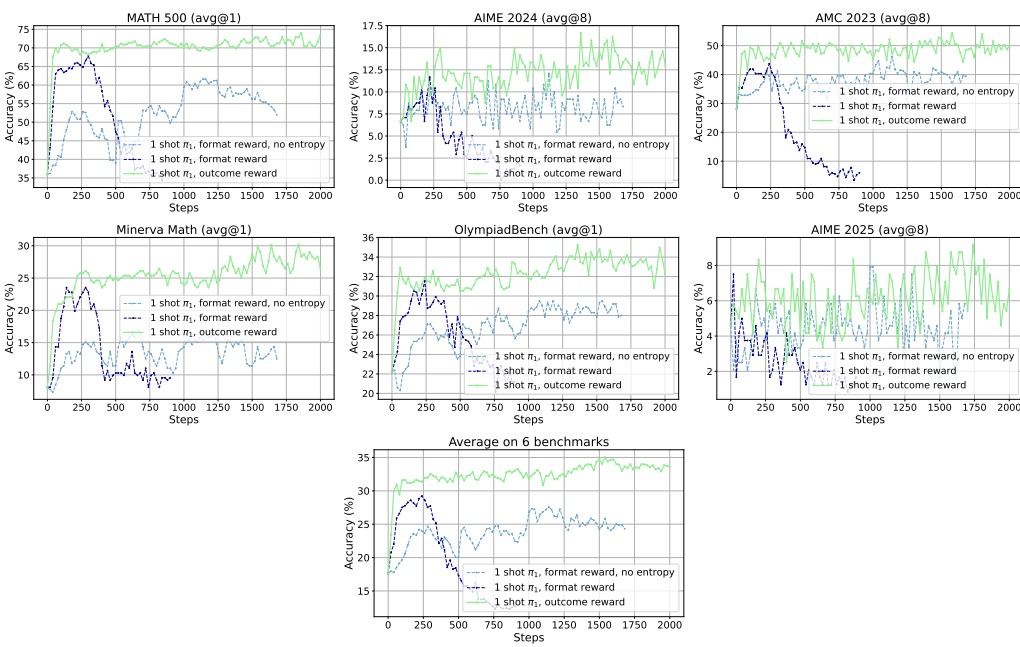

Figure 14: **Comparison between outcome reward and format reward for 1-shot RLVR with $\pi_1$ on Qwen2.5-Math-1.5B.**

Table 14: **RLVR with only format reward can still improve model performance significantly, while still having a gap compared with that using outcome reward.** Numbers with orange color denote the ratio of responses that contain "\boxed{}" in evaluation. Here we consider adding entropy loss or not for format reward. Detailed test curves are in Fig. 13 and Fig. 14. We can see that: **(1)** RLVR with format reward has similar test performance between 1.2k dataset DSR-sub and $\pi_1$. **(2)** $\pi_1$ with outcome reward or format reward have similar \boxed{} ratios, but the former still has better test performance (e.g., +7.4% on MATH500 and +5.8% on average). **(3)** Interestingly, RLVR with DSR-sub using outcome reward can fix the format perfectly, although it still has similar test performance as 1-shot RLVR with $\pi_1$ (outcome reward).

| Dataset | Reward Type | Entropy Loss | MATH 500 | AIME 2024 | AMC 2023 | Minerva Math | Olympiad-Bench | AIME 2025 | Avg. |
|---|---|---|---|---|---|---|---|---|---|
| NA | NA | NA | $36.0_{60\%}$ | $6.7_{75\%}$ | $28.1_{83\%}$ | $8.1_{59\%}$ | $22.2_{76\%}$ | $4.6_{81\%}$ | $17.6_{72\%}$ |
| DSR-sub | Outcome | + | $\mathbf{73.6}_{100\%}$ | $\mathbf{17.1}_{99\%}$ | $\mathbf{50.6}_{100\%}$ | $\mathbf{32.4}_{99\%}$ | $\mathbf{33.6}_{99\%}$ | $\mathbf{8.3}_{100\%}$ | $\mathbf{35.9}_{99\%}$ |
| DSR-sub | Format | + | $65.0_{94\%}$ | $8.3_{83\%}$ | $45.9_{94\%}$ | $17.6_{89\%}$ | $29.9_{92\%}$ | $5.4_{90\%}$ | $28.7_{91\%}$ |
| DSR-sub | Format | | $61.4_{93\%}$ | $9.6_{87\%}$ | $44.7_{94\%}$ | $16.5_{83\%}$ | $29.5_{90\%}$ | $3.8_{87\%}$ | $27.6_{89\%}$ |
| $\{\pi_1\}$ | Outcome | + | $\mathbf{72.8}_{97\%}$ | $\mathbf{15.4}_{92\%}$ | $51.6_{97\%}$ | $29.8_{92\%}$ | $33.5_{88\%}$ | $7.1_{93\%}$ | $\mathbf{35.0}_{93\%}$ |
| $\{\pi_1\}$ | Outcome | | $68.2_{97\%}$ | $\mathbf{15.4}_{92\%}$ | $49.4_{95\%}$ | $25.0_{94\%}$ | $31.7_{91\%}$ | $5.8_{90\%}$ | $32.6_{93\%}$ |
| $\{\pi_1\}$ | Format | + | $65.4_{96\%}$ | $8.8_{91\%}$ | $43.8_{98\%}$ | $22.1_{91\%}$ | $31.6_{90\%}$ | $3.8_{88\%}$ | $29.2_{92\%}$ |
| $\{\pi_1\}$ | Format | | $61.6_{92\%}$ | $8.3_{84\%}$ | $46.2_{90\%}$ | $15.4_{78\%}$ | $29.3_{89\%}$ | $4.6_{86\%}$ | $27.6_{88\%}$ |

**significantly** (e.g., about 29% improvement on MATH500 and about 11% gain on average). **(3) There is still significant gap between the performance of 1-shot RLVR with outcome reward using $\pi_1$ and that of format-reward RLVR** (e.g., +7.4% on MATH500 and +5.8% on average), although they may have similar ratios of responses that contain "\boxed{}" in evaluation (More discussions are in (b) part). **(4)** In particular, format-reward RLVR is more sensitive to entropy loss based on Fig. 14 and Fig. 13.

Interestingly, we also note that the best performance of format-reward RLVR on MATH500 and AIME24 are close to that for 1-shot RLVR with relatively worse examples, for example, $\pi_7$ and $\pi_{11}$ in Tab. 3. This may imply that *1-shot RLVR with outcome reward can at least work as well as format-reward RLVR, but with proper examples that can better incentivize the reasoning capability of the model, 1-shot RLVR with outcome reward can bring additional non-trivial improvement.* Appendix C.2.5 provides a prompt $\pi_1'$, which uses a sub-question of $\pi_1$, as an example to support our claim here.

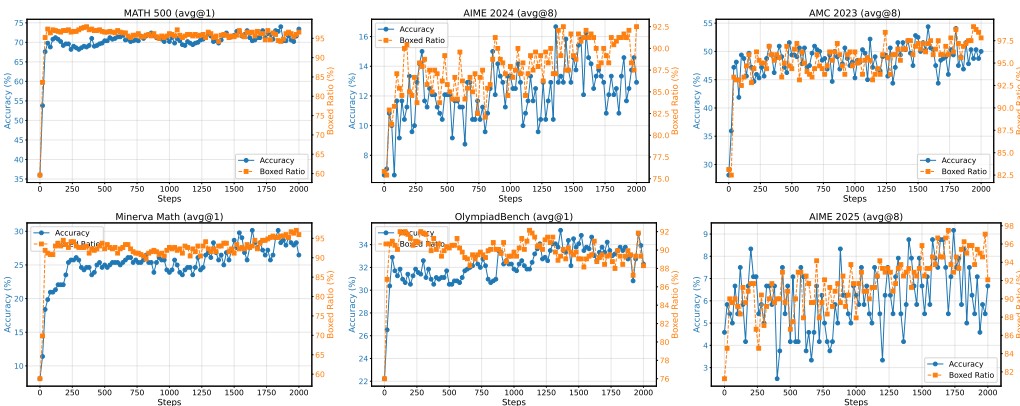

Figure 15: **Relation between the number of** \boxed{} **and test accuracy**. We can see that they have a strong positive correlation. However, after the number of \boxed{} enters a plateau, the evaluation results on some evaluation tasks continue improving (like Minerva Math, OlympiadBench and MATH500).

**(b) Observe the change of format in 1-shot RLVR.** We then investigate how the output format of the model, for example, the number of \boxed{}, changes in the 1-shot RLVR progress. The results are shown in Fig. 15. We can see that (1) the test accuracy is strongly positively correlated to the number of \boxed{}, which matches our claim that format fixing contributes a lot to model

Table 15: **1-shot RLVR does not do something like put the answer into the** `\boxed{}`**.** "Ratio of disagreement" means the ratio of questions that has different judgement between Qwen-Eval and QwQ-32B judge. Here we let QwQ-32B judged based on if the output contain correct answer, without considering if the answer is put in the `\boxed{}`.

| | Step0 | Step 20 | Step 60 | Step 500 | Step 1300 | Step 1860 |
|---|---|---|---|---|---|---|
| Ratio of `\boxed{}` | 59.6% | 83.6% | 97.4% | 96.6% | 96.6% | 94.2% |
| Acc. judge by Qwen-Eval | 36.0 | 53.8 | 69.8 | 70.4 | 72.2 | 74.0 |
| Acc. judge by QwQ-32B | 35.8 | 57.2 | 70.6 | 71.8 | 73.6 | 74.6 |
| Ratio of disagreement | 4.2% | 5% | 1.2% | 1.4% | 1.8% | 1.8% |

Table 16: $\pi_1$ **even performs well for in-context learning on Qwen2.5-Math-7B.** Here "Qwen official 4 examples" are from Qwen Evaluation repository [25] for 4-shot in-context learning on MATH500, and "Qwen official Example 1" is the first example.

| Dataset | Method | MATH 500 | AIME 2024 | AMC 2023 | Minerva Math | Olympiad-Bench | AIME 2025 | Avg. |
|---|---|---|---|---|---|---|---|---|
| **Qwen2.5-Math-1.5B** | | | | | | | | |
| NA | NA | 36.0 | 6.7 | 28.1 | 8.1 | 22.2 | 4.6 | 17.6 |
| $\{\pi_1\}$ | RLVR | 72.8 | **15.4** | **51.6** | **29.8** | **33.5** | **7.1** | **35.0** |
| $\{\pi_1\}$ | In-Context | 59.0 | 8.3 | 34.7 | 19.9 | 25.6 | 5.4 | 25.5 |
| Qwen official 4 examples | In-Context | 49.8 | 1.7 | 16.9 | 19.9 | 19.9 | 0.0 | 18.0 |
| Qwen official Example 1 | In-Context | 34.6 | 2.5 | 14.4 | 12.1 | 21.0 | 0.8 | 14.2 |
| **Qwen2.5-Math-7B** | | | | | | | | |
| NA | NA | 51.0 | 12.1 | 35.3 | 11.0 | 18.2 | 6.7 | 22.4 |
| $\{\pi_1\}$ | RLVR | **79.2** | **23.8** | **60.3** | 27.9 | 39.1 | 10.8 | **40.2** |
| $\{\pi_1\}$ | In-Context | 75.4 | 15.8 | 48.4 | **30.1** | **41.3** | **13.3** | 37.4 |
| Qwen official 4 examples | In-Context | 59.2 | 4.2 | 20.9 | 20.6 | 24.4 | 0.8 | 21.7 |
| Qwen official Example 1 | In-Context | 54.0 | 4.2 | 23.4 | 18.4 | 21.2 | 2.1 | 20.6 |

improvement in (a), but (2) for some benchmarks like MATH500, Minerva Math and OlympiadBench, when the number of `\boxed{}` keeps a relatively high ratio, the test accuracy on these benchmarks is still improving, which may imply independent improvement of reasoning capability.

In particular, to prevent the case that the model outputs the correct answer but not in `\boxed{}`, we also use LLM-as-a-judge [61] with QwQ-32B [62] to judge if the model contains the correct answer in the response. The results are shown in Tab. 15. We can see that the accuracy judged by rule-based Qwen-Eval pipeline and LLM judger QwQ-32B are very close, and as the ratio of `\boxed{}` increases, the test accuracy also increases, which implies that the number of correct answers exhibited in the response also increases, rather than just putting correct answer into `\boxed{}`.

Notably, we also observe that Qwen2.5-Math models contain lots of repetition at the end of model responses, which may result in failure of obtaining final results. The ratio of repetition when evaluating MATH500 can be as high as about 40% and 20% for Qwen2.5-Math-1.5B and Qwen2.5-Math-7B, respectively, which is only about 2% for Llama3.2-3B-Instruct. This may result in the large improvement of format fixing (e.g., format-reward RLVR) mentioned in (a).

**(c) In-context learning with one-shot example.** In-context learning [63] is a widely-used baseline for instruction following (although it may still improve model's reasoning capability). In this section, we try to see if 1-shot RLVR can behave better than in-context learning. Especially, we consider the official 4 examples chosen by Qwen-Eval [25] for in-context learning, and also the single training example $\pi_1$. The results are shown in Tab. 16.

We can find that (1) surprisingly, $\pi_1$ **with self-generated response can behave much better than Qwen's official examples**, both for 1.5B and 7B models. In particular on Qwen2.5-Math-7B, in-context learning with $\pi_1$ can improve MATH500 from 51.0% to 75.4% and on average from 22.4% to 37.4%. (2) Although in-context learning also improves the base models, 1-shot RLVR still performs better than all in-context results, showing the advantage of RLVR.

Table 17: **Influence of Random Wrong Labels.** Here "Error Rate" means the ratio of data that has the random wrong labels.

| Dataset | Error Rate | MATH 500 | AIME 2024 | AMC 2023 | Minerva Math | Olympiad-Bench | AIME 2025 | Avg. |
|---|---|---|---|---|---|---|---|---|
| NA | NA | 36.0 | 6.7 | 28.1 | 8.1 | 22.2 | 4.6 | 17.6 |
| **Qwen2.5-Math-1.5B + GRPO** | | | | | | | | |
| DSR-sub | 0% | 73.6 | 17.1 | 50.6 | 32.4 | 33.6 | 8.3 | 35.9 |
| DSR-sub | 60% | 71.8 | 17.1 | 47.8 | 29.4 | 34.4 | 7.1 | 34.6 |
| DSR-sub | 90% | 67.8 | 14.6 | 46.2 | 21.0 | 32.3 | 5.4 | 31.2 |
| $\{\pi_1\}$ | 0% | 72.8 | 15.4 | 51.6 | 29.8 | 33.5 | 7.1 | 35.0 |
| **Qwen2.5-Math-1.5B + PPO** | | | | | | | | |
| DSR-sub | 0% | 72.8 | 19.2 | 48.1 | 27.9 | 35.0 | 9.6 | 35.4 |
| DSR-sub | 60% | 71.6 | 13.3 | 49.1 | 27.2 | 34.4 | 12.1 | 34.6 |
| DSR-sub | 90% | 68.2 | 15.8 | 50.9 | 26.1 | 31.9 | 4.6 | 32.9 |
| $\{\pi_1\}$ | 0% | 72.4 | 11.7 | 51.6 | 26.8 | 33.3 | 7.1 | 33.8 |

In short, we use these three methods to confirm that 1-shot RLVR indeed does format fixing and obtains a lot of gain from it, but it still has additional improvement that cannot be easily obtained from format reward or in-context learning.

### C.2.4 Influence of Random Wrong Labels

In this section, we want to investigate the label robustness of RLVR. It's well-known that general deep learning is robust to label noise [64], and we want to see if this holds for RLVR. We try to randomly flip the labels of final answers in DSR-sub and see their performance. Here we randomly add or subtract numbers within 10 and randomly change the sign. If it is a fraction, we similarly randomly add or subtract the numerator and denominator.

The results are in Tab. 17. We can see that (1) changing 60% of the data with wrong labels can still achieve good RLVR results. (2) if 90% of the data in the dataset contains wrong labels (i.e., only about 120 data contain correct labels, and all other 1.1k data have wrong labels), the model performance will be worse than that for 1-shot RLVR with $\pi_1$ (which only contains 1 correct label!). This may show that RLVR is partially robust to label noise, but if there are too many data with random wrong labels, they may hurt the improvement brought by data with correct labels.

### C.2.5 Change the Prompt of $\pi_1$

Table 18: **Keeping CoT complexity in problem-solving may improve model performance.** Comparing $\pi_1$ and simplified variant $\pi_1'$ (prompt: "Calculate $\sqrt[3]{2048}$"), where we only keep the main step that Qwen2.5-Math-1.5B may make a mistake on. We record the results from the checkpoint with the best average performance. For $\pi_1'$, the model's output CoT is simpler and the corresponding 1-shot RLVR performance is worse. The additional improvement of $\pi_1'$ is relatively marginal compared with using format reward, showing the importance of the training example used in 1-shot RLVR.

| RL Dataset | Reward Type | MATH 500 | AIME 2024 | AMC 2023 | Minerva Math | Olympiad-Bench | AIME 2025 | Avg. |
|---|---|---|---|---|---|---|---|---|
| **Qwen2.5-Math-1.5B [24]** | | | | | | | | |
| NA | NA | 36.0 | 6.7 | 28.1 | 8.1 | 22.2 | 4.6 | 17.6 |
| $\{\pi_1\}$ | outcome | **72.8** | **15.4** | **51.6** | **29.8** | **33.5** | **7.1** | **35.0** |
| Simplified $\{\pi_1'\}$ | outcome | 65.4 | 9.6 | 45.9 | 23.2 | 31.1 | 5.0 | 30.0 |
| DSR-sub | Format | 65.0 | 8.3 | 45.9 | 17.6 | 29.9 | 5.4 | 28.7 |

As discussed in Sec. 3.2.1, we show that the model can almost solve $\pi_1$ but sometimes fails in solving its last step: "Calculate $\sqrt[3]{2048}$". We use this step itself as a problem ($\pi_1'$), and see how it behaves in 1-shot RLVR. The results are in Tab. 18. Interestingly, we find that $\pi_1'$ significantly underperforms $\pi_1$ and has only 1.3% average improvement compared with format reward (as illustrated in

Appendix C.2.3 (a)). We think the reason should be that although solving $\sqrt[3]{2048}$ is one of the most difficult parts of $\pi_1$, $\pi_1$ still needs other key steps to solve (e.g., calculating $k$ from $P = kAV^3$ given some values) that may generate different patterns of CoT (rather than just calculating), which may allow more exploration space at the post-saturation generalization stage and maybe better incentivize the model's reasoning capability.

## C.3 Response Length

In Tab. 19, we report the average response length on the evaluation tasks. The response length on the test tasks remains relatively stable compared to that on the training data.

Table 19: **Average response length of Qwen2.5-Math-1.5B on evaluation tasks.** We use the format-reward experiment (DSR-sub + format reward in Tab. 14) as the baseline to eliminate differences in token counts introduced by formats.

| Setting | MATH 500 | AIME24 2024 | AMC23 2023 | Minerva Math | Olympiad-Bench | AIME 2025 | Avg. |
|---|---|---|---|---|---|---|---|
| Format Reward | 689 | 1280 | 911 | 1018 | 957 | 1177 | 1005 |
| 1-shot RLVR w/ $\pi_1$ (step 100) | 611 | 1123 | 939 | 1072 | 951 | 1173 | 978 |
| 1-shot RLVR w/ $\pi_1$ (step 1500) | 740 | 1352 | 986 | 905 | 1089 | 1251 | 1054 |
| RLVR w/ DSR-sub (step 100) | 636 | 1268 | 874 | 797 | 954 | 1122 | 942 |
| RLVR w/ DSR-sub (step 1500) | 562 | 949 | 762 | 638 | 784 | 988 | 780 |

## C.4 `Pass@8` **Results**

In Tab. 20, we report the `pass@8` results on the evaluation tasks. Interestingly, we find that (1) 1-shot RLVR achieves comparable or even slightly better `pass@8` performance (51.7(2) full-set RLVR (with 1.2k DSR-sub) exhibits a noticeable downward trend in `pass@8` performance after 200 steps, which is consistent with recent findings that RLVR may sometimes degrade the `pass@n` performance [20].

Table 20: `Pass@8` **results on 3 math evaluation tasks using Qwen2.5-Math-1.5B.** We also include the performance of RLVR with format-reward (as in Table 19) as a stronger baseline.

| Setting | AIME24 | AIME25 | AMC23 | Avg. (3 tasks) |
|---|---|---|---|---|
| Base Model | 26.6 | 20.0 | 72.5 | 39.7 |
| Format Reward(highest) | 33.3 | 23.3 | 72.5 | 43.1 |
| RLVR w/ DSR-sub (highest, step 160) | 36.7 | 26.7 | 87.5 | 50.3 |
| RLVR w/ DSR-sub (step 500) | 33.3 | 30.0 | 82.5 | 48.6 |
| RLVR w/ DSR-sub (step 1000) | 33.3 | 20.0 | 75.0 | 42.8 |
| RLVR w/ DSR-sub (step 1500) | 30.0 | 26.7 | 67.5 | 41.3 |
| 1-shot RLVR (step 500) | 30.0 | 16.7 | 80.0 | 42.2 |
| 1-shot RLVR (highest, step 980) | 36.7 | 33.3 | 85.0 | 51.7 |
| 1-shot RLVR (step 1500) | 26.6 | 23.3 | 87.5 | 45.8 |

# D Discussions

## D.1 Limitations of Our Work

Due to the limit of computational resources, we haven't tried larger models like Qwen2.5-32B training currently. But in general, a lot of RLVR works are conducted on 1.5B and 7B models, and they already achieve impressive improvement on some challenging math benchmarks like OlympiadBench, so our experiments are still insightful for RLVR topics. Another limitation of our work is that we mainly focus on the math domain, but haven't tried 1(few)-shot RLVR on other verifiable domains like coding. But we also emphasize that all math-related experiments and conclusions in our paper are logically self-contained and clearly recorded, to ensure clarity and avoid confusion for readers. And we mainly focus on analyzing this new phenomenon itself, which already brings a lot of novel observations (e.g., cross-category generalization, post-saturation generalization, and more frequent self-reflection in 1-shot RLVR, etc.). We leave the few-shot RLVR on other scenarios for future work.

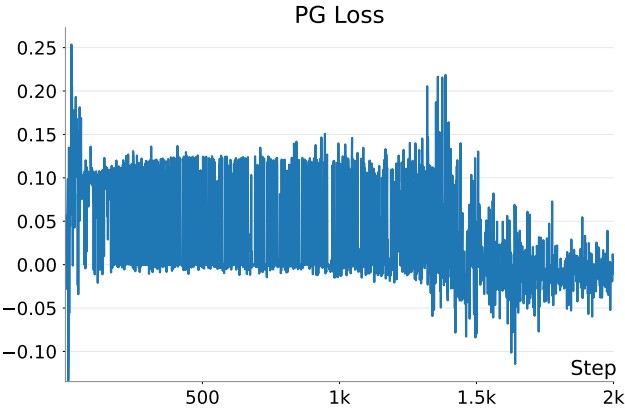

Figure 16: **The norm of policy gradient loss for 1-shot RLVR ($\pi_1$) on Qwen2.5-Math-1.5B.**

In particular, we note that our main focus is to propose a new observation rather than propose a new better method, noting that 1-shot RLVR doesn't save (and maybe requires more) RL computation. Besides, $\pi_1$ is not necessarily the best choice for 1-shot RLVR on other models, since it's selected based on the historical variance score of Qwen2.5-Math-1.5B. In general, using few-shot RLVR may be more stable for training, as we have seen that on DeepSeek-R1-Distill-Qwen-1.5B (Tab. 4), Qwen2.5-Math-7B (Tab. 4, 10) and Qwen2.5-1.5B (Tab. 10), RLVR with 16 examples ($\{\pi_1, \ldots, \pi_{16}\}$) works as well as RLVR with 1.2k dataset DSR-sub and outperforms 1-shot RL with $\pi_1$.

### D.2 Reasoning Capability of Base Models

The effectiveness of 1(few)-shot RLVR provides strong evidence for an assumption people proposed recently, that is, base models already have strong reasoning capability [13, 6, 20, 21]. For example, Dr. GRPO [13] has demonstrated that when no template is used, base models can achieve significantly better downstream performance. Recent work further supports this observation by showing that, with respect to the pass@k metrics, models trained via RLVR gradually perform worse than the base model as $k$ increases [20]. Our work corroborates this claim from another perspective, as a single example provides almost no additional knowledge. Moreover, our experiments reveal that using very few examples with RLVR is already sufficient to achieve significant improvement on mathematical reasoning tasks. Thus, it is worth investigating how to select appropriate data to better activate the model during the RL stage *while maintaining data efficiency*.

### D.3 Why Model Continues Improving After the Training Accuracy Reaches Near 100%?

A natural concern of 1-shot RLVR is that if training accuracy reaches near 100% (which may occur when over-training on one example), the GRPO advantage (Eqn. 6) should be zero, eliminating policy gradient signal. However, entropy loss encourages diverse outputs, causing occasional errors ( 99.x% training accuracy) and non-zero gradients (advantage becomes large for batches with wrong responses due to small variance). This shows the importance of entropy loss to the post-saturation generalization (Fig. 5). Supporting this, Fig. 16 shows that for 1-shot RLVR training ($\pi_1$) on Qwen2.5-Math-1.5B, policy gradient loss remains non-zero after 100 steps.

### D.4 Future Works

We believe our findings can provide some insights for the following topics:

**Data Selection and Curation.** Currently, there are no specific data selection methods for RLVR except LIMR [19]. Note that 1-shot RLVR allows for evaluating each example individually, it will be helpful for assessing the data value, and thus help to design better data selection strategy. What's more, noting that different examples can have large differences in stimulating LLM reasoning capability (Tab. 3), it may be necessary to find out what kind of data is more useful for RLVR, which is critical for the RLVR data collection stage. It's worth mentioning that **our work does not mean scaling RLVR datasets is useless**, but it emphasizes the importance of better selection and collection of data for RLVR.

Table 21: **Details of example $\pi_{13}$.**

| Prompt |
| --- |
| Given that circle $C$ passes through points $P(0,-4)$, $Q(2,0)$, and $R(3,-1)$. \n$(1)$ Find the equation of circle $C$. \n$(2)$ If the line $l$: mx+y-1=0$ intersects circle $C$ at points $A$ and $B$, and $\|AB\|=4$, find the value of $m$. Let's think step by step and output the final answer within \\boxed{}. |
| **Ground truth (label in DSR-sub):** $\frac{4}{3}$. |

Table 22: **Details of example $\pi_2$.**

| Prompt: |
| --- |
| How many positive divisors do 9240 and 13860 have in common? Let's think step by step and output the final answer within \\boxed{}. |
| **Ground truth (label in DSR-sub):** 24. |

**Understanding 1-shot RLVR and Post-saturation Generalization**   A rigorous understanding for the feasibility of 1-shot LLM RLVR and post-saturation generalization is still unclear. We think that one possible hypothesis is that the policy loss on the learned examples plays a role as "implicit regularization" of RLVR when the model tries to explore more diverse output strategies under the encouragement of entropy loss or larger rollout temperature. It will punish the exploration patterns that make the model fail to answer the learned data, and thus provide a verification for exploration. It's interesting to explore if the phenomenon has relevance to Double Descent [65] or the implicit regularization from SGD [66, 67], as 1-shot RLVR on $\pi_{13}$ (Fig. 2, middle) shows a test curve similar to Double Descent. We leave the rigorous analysis of this phenomenon for future works, and we believe that can help us to comprehend what happens in the RLVR process.

**Importance of Exploration.**   In Sec. 4.1, we also highlight the importance of entropy loss in 1-shot RLVR, and note that a more thorough explanation of why training with only entropy loss can enhance model performance remains an interesting direction for future work (Sec. 4.2). Relatedly, entropy loss has also received increasing attention from the community, with recent works discussing its dynamics [68, 47, 60] or proposing improved algorithms from the perspective of entropy [46]. Moreover, we believe a broader and more important insight for these is that encouraging the model to explore more diverse outputs within the solution space is critical, as it may significantly impact the model's generalization to downstream tasks [69]. Adding entropy loss is merely one possible approach to achieve this goal and may not necessarily be the optimal solution. As shown in our paper and previous work [60], the effectiveness of entropy loss is sensitive to the choice of coefficient, which could limit its applicability in larger-scale experiments. We believe that discovering better strategies to promote exploration could further enhance the effectiveness of RLVR.

**Other Applications.**   In this paper, we focus primarily on mathematical reasoning data; however, it is also important to evaluate the efficacy of 1-shot RLVR in other domains, such as code generation or tasks without verifiable rewards. Moreover, investigating methodologies to further improve few-shot RLVR performance under diverse data-constrained scenarios represents a valuable direction. Examining the label robustness of RLVR, as discussed in Sec. 4.2, likewise merits further exploration. Finally, these observations may motivate the development of additional evaluation sets to better assess differences between 1-shot and full-set RLVR on mathematical or other reasoning tasks.

# E   Example Details

In the main paper, we show the details of $\pi_1$. Another useful example $\pi_{13}$ is shown in Tab. 21. Here we mention that $\pi_{13}$ is a geometry problem and its answer is precise. And similar to $\pi_1$, the initial base model still has 21.9% of outputs successfully obtaining $\frac{4}{3}$ in 128 samplings.

Besides, Tab. 22 through 42 in the supplementary material provide detailed information for each example used in our experiments and for all other examples in $\{\pi_1, \ldots, \pi_{17}\}$. Each table contains the specific prompt and corresponding ground truth label for an individual example.

Table 23: **Details of example** $\pi_3$.

**Prompt:**

There are 10 people who want to choose a committee of 5 people among them. They do this by first electing a set of $1,2,3$, or 4 committee leaders, who then choose among the remaining people to complete the 5-person committee. In how many ways can the committee be formed, assuming that people are distinguishable? (Two committees that have the same members but different sets of leaders are considered to be distinct.) Let's think step by step and output the final answer within \\boxed{}.

**Ground truth (label in DSR-sub):** 7560.

Table 24: **Details of example** $\pi_4$.

**Prompt:**

Three integers from the list $1,2,4,8,16,20$ have a product of 80. What is the sum of these three integers? Let's think step by step and output the final answer within \\boxed{}.

**Ground truth (label in DSR-sub):** 25.

Table 25: **Details of example** $\pi_5$.

**Prompt:**

In how many ways can we enter numbers from the set $\\{1,2,3,4\\}$ into a $4 \\times 4$ array so that all of the following conditions hold? (a) Each row contains all four numbers. (b) Each column contains all four numbers. (c) Each "quadrant" contains all four numbers. (The quadrants are the four corner $2 \\times 2$ squares.) Let\'s think step by step and output the final answer within \\boxed{}.

**Ground truth (label in DSR-sub):** 288.

Table 26: **Details of example** $\pi_6$.

**Prompt:**

The vertices of a $3 \\times 1 \\times 1$ rectangular prism are $A, B, C, D, E, F, G$, and $H$ so that $A E, B F$, $C G$, and $D H$ are edges of length 3. Point $I$ and point $J$ are on $A E$ so that $A I=I J=J E=1$. Similarly, points $K$ and $L$ are on $B F$ so that $B K=K L=L F=1$, points $M$ and $N$ are on $C G$ so that $C M=M N=N G=1$, and points $O$ and $P$ are on $D H$ so that $D O=O P=P H=1$. For every pair of the 16 points $A$ through $P$, Maria computes the distance between them and lists the 120 distances. How many of these 120 distances are equal to $\\sqrt{2}$? Let's think step by step and output the final answer within \\boxed{}.

**Ground truth (label in DSR-sub):** 32.

Table 27: **Details of example** $\pi_7$.

**Prompt:**

Set $u_0 = \\frac{1}{4}$, and for $k \\ge 0$ let $u_{k+1}$ be determined by the recurrence\n \\[u_{k+1} = 2u_k - 2u_k^2.\\]This sequence tends to a limit; call it $L$. What is the least value of $k$ such that\n\\[|u_k-L| \\le \\frac{1}{2^{1000}}?\\] Let's think step by step and output the final answer within \\boxed{}.

**Ground truth (label in DSR-sub):** 10.

Table 28: **Details of example** $\pi_8$.

**Prompt:**

Consider the set $\\{2, 7, 12, 17, 22, 27, 32\\}$. Calculate the number of different integers that can be expressed as the sum of three distinct members of this set. Let's think step by step and output the final answer within \\boxed{}.

**Ground truth (label in DSR-sub):** 13.

Table 29: **Details of example** $\pi_9$.

**Prompt:**

In a group photo, 4 boys and 3 girls are to stand in a row such that no two boys or two girls stand next to each other. How many different arrangements are possible? Let's think step by step and output the final answer within \\boxed{}.

**Ground truth (label in DSR-sub):** 144.

Table 30: **Details of example** $\pi_{10}$.

**Prompt:**

How many ten-digit numbers exist in which there are at least two identical digits? Let's think step by step and output the final answer within \\boxed{}.

**Ground truth (label in DSR-sub):** 8996734080.

Table 31: **Details of example** $\pi_{11}$.

**Prompt:**

How many pairs of integers $a$ and $b$ are there such that $a$ and $b$ are between $1$ and $42$ and $a^9 = b^7 \\mod 43$ ? Let's think step by step and output the final answer within \\boxed{}.

**Ground truth (label in DSR-sub):** 42.

Table 32: **Details of example** $\pi_{12}$.

**Prompt:**

Two springs with stiffnesses of $6 \\, \\text{kN} / \\text{m}$ and $12 \\, \\text{kN} / \\text{m}$ are connected in series. How much work is required to stretch this system by 10 cm? Let's think step by step and output the final answer within \\boxed{}.

**Ground truth (label in DSR-sub):** 20.

Table 33: **Details of example** $\pi_{14}$.

**Prompt:**

Seven cards numbered $1$ through $7$ are to be lined up in a row. Find the number of arrangements of these seven cards where one of the cards can be removed leaving the remaining six cards in either ascending or descending order. Let's think step by step and output the final answer within \\boxed{}.

**Ground truth (label in DSR-sub):** 74.

Table 34: **Details of example** $\pi_{15}$.

**Prompt:**

What is the area enclosed by the geoboard quadrilateral below?\n[asy] unitsize(3mm); defaultpen(linewidth(.8pt)); dotfactor=2;  for(int a=0; a<=10; ++a) for(int b=0; b<=10; ++b) {   dot((a,b));  };  draw((4,0)--(0,5)--(3,4)--(10,10)--cycle); [/asy] Let's think step by step and output the final answer within \\boxed{}.

**Ground truth (label in DSR-sub):** $22\frac{1}{2}$.

Table 35: **Details of example** $\pi_{16}$.

**Prompt:**

If $p, q,$ and $r$ are three non-zero integers such that $p + q + r = 26$ and\\[\\frac{1}{p} + \\frac{1}{q} + \\frac{1}{r} + \\frac{360}{pqr} = 1,\\] compute $pqr$.\n Let's think step by step and output the final answer within \\boxed{}.

**Ground truth (label in DSR-sub):** 576.

Table 36: **Details of example** $\pi_{17}$.

**Prompt:**

In Class 3 (1), consisting of 45 students, all students participate in the tug-of-war. For the other three events, each student participates in at least one event. It is known that 39 students participate in the shuttlecock kicking competition and 28 students participate in the basketball shooting competition. How many students participate in all three events? Let's think step by step and output the final answer within \\boxed{}.

**Ground truth (label in DSR-sub):** 22.

Table 37: **Details of example** $\pi_{605}$.

**Prompt:**

Given vectors $$\\overrightarrow {m}=( \\sqrt {3}\\sin x+\\cos x,1), \\overrightarrow {n}=(\\cos x,-f(x)),
\\overrightarrow {m}\\perp \\overrightarrow {n}$$.\n(1) Find the monotonic intervals of $f(x)$;\n(2) Given
that $A$ is an internal angle of $\\triangle ABC$, and $$f\\left( \\frac {A}{2}\\right)= \\frac {1}{2}+
\\frac { \\sqrt {3}}{2},a=1,b= \\sqrt {2}$$, find the area of $\\triangle ABC$. Let's think step by step
and output the final answer within \\boxed{}.

**Ground truth (label in DSR-sub):** $\frac{\sqrt{3}-1}{4}$.

Table 38: **Details of example** $\pi_{606}$.

**Prompt:**

How many zeros are at the end of the product \\( s(1) \\cdot s(2) \\cdot \\ldots \\cdot s(100) \\), where
\\( s(n) \\) denotes the sum of the digits of the natural number \\( n \\)? Let's think step by step and
output the final answer within \\boxed{}.

**Ground truth (label in DSR-sub):** 19.

Table 39: **Details of example** $\pi_{1201}$.

**Prompt:**

The angles of quadrilateral $PQRS$ satisfy $\\angle P = 3\\angle Q = 4\\angle R = 6\\angle S$. What is the
degree measure of $\\angle P$? Let's think step by step and output the final answer within \\boxed{}.

**Ground truth (label in DSR-sub):** 206.

Table 40: **Details of example** $\pi_{1207}$. A correct answer for this question should be $2/3$.

**Prompt:**

A rectangular piece of paper whose length is $\\sqrt{3}$ times the width has area $A$. The paper is divided
into three equal sections along the opposite lengths, and then a dotted line is drawn from the first divider
to the second divider on the opposite side as shown. The paper is then folded flat along this dotted line to
create a new shape with area $B$. What is the ratio $\\frac{B}{A}$? Let's think step by step and output the
final answer within \\boxed{}.

**Ground truth (label in DSR-sub):** $\frac{4}{5}$.

Table 41: **Details of example** $\pi_{1208}$.

**Prompt:**

Given a quadratic function in terms of \\\\(x\\\\), \\\\(f(x)=ax^{2}-4bx+1\\\\).\n\\\\((1)\\\\) Let set
\\\\(P=\\\\{1,2,3\\\\}\\\\) and \\\\(Q=\\\\{-1,1,2,3,4\\\\}\\\\), randomly pick a number from set
\\\\(P\\\\) as \\\\(a\\\\) and from set \\\\(Q\\\\) as \\\\(b\\\\), calculate the probability that the
function \\\\(y=f(x)\\\\) is increasing in the interval \\\\([1,+\\\\infty)\\\\).\n\\\\((2)\\\\) Suppose point
\\\\((a,b)\\\\) is a random point within the region defined by \\\\( \\\\begin{cases} x+y-8\\\\leqslant 0
\\\\\\\\ x > 0 \\\\\\\\ y > 0\\\\end{cases}\\\\), denote \\\\(A=\\\\{y=f(x)\\\\) has two zeros, one greater
than \\\\(1\\\\) and the other less than \\\\(1\\\\}\\\\), calculate the probability of event \\\\(A\\\\)
occurring. Let's think step by step and output the final answer within \\boxed{}.

**Ground truth (label in DSR-sub):** $\frac{961}{1280}$.

Table 42: **Details of example** $\pi_{1209}$.

**Prompt:**

Define the derivative of the $(n-1)$th derivative as the $n$th derivative $(n \\in N^{*}, n \\geqslant 2)$,
that is, $f^{(n)}(x)=[f^{(n-1)}(x)]'$. They are denoted as $f''(x)$, $f'''(x)$, $f^{(4)}(x)$, ...,
$f^{(n)}(x)$. If $f(x) = xe^{x}$, then the $2023$rd derivative of the function $f(x)$ at the point
$(0, f^{(2023)}(0))$ has a $y$-intercept on the $x$-axis of ______. Let's think step by step and output the
final answer within \\boxed{}.

**Ground truth (label in DSR-sub):** $-\frac{2023}{2024}$.

