# OpenReview forum: "Reinforcement Learning for Reasoning in Large Language Models with One Training Example"
_NeurIPS.cc/2025/Conference — NeurIPS 2025 poster_

### Official Review · Reviewer_URox · 2025-06-30

**Clarity:** 3
**Significance:** 3
**Originality:** 3
**Rating:** 5
**Confidence:** 4

**Summary:**

The paper found that one training sample is sufficient to lead to substantive performance gains in RLVR. It discusses methods for selecting the sample, conducts experiments on different samples and models, and concludes that one or a few training samples can bring performance gains similar to training on the full dataset.

**Questions:**

1. In Figure 4, I observe that the response length of training on a single sample is much longer than on the full dataset. Would the response length in the evaluation dataset also be substantially longer? If so, this may be a significant drawback of training on a single sample that needs to be emphasized, due to the increased inference cost (even if the performance is similar).

2. The evaluation mainly uses pass@1 with temp=0 and pass@1 with temp=0.6. I am interested in how pass@n (e.g., n=8, with a decent temperature such as 0.6) compares between training on a full dataset and a single sample. Given the large entropy shown in Figure 4 for 1-shot training, will pass@n be worse than when training on a full dataset?

**Ethical Concerns:**

["NO or VERY MINOR ethics concerns only"]

**Final Justification:**

I recommend acceptance, as the authors have adequately addressed my questions and the work provides novel insights into how RLVR operates in LLMs.

**Limitations:**

Limitations are addressed.

**Paper Formatting Concerns:**

None.

**Quality:**

3

**Strengths And Weaknesses:**

**Strengths**

- Novel finding: The phenomenon that one training sample can work with RLVR is surprising and counter-intuitive and should be investigated further by the research community.
- Extensive experiments: Multiple models and different training samples are used, strengthening the claim that this phenomenon is not specific to a particular model or a particular training sample. The ablation analysis is comprehensive, such as including ablations on different components of the training loss.

**Weaknesses**

- A more in-depth analysis could strengthen the paper. Section 4.1 discusses that the underlying reason for the phenomenon is not due to grokking, while Section 4.2 shows that entropy alone may also help in the performance gain. However, neither section analyzes the underlying reason why a single sample is sufficient.
- Slightly misleading narrative in Section 3.2.1: In Section 3.2.1, it makes the observation that "The answer is not precise"—i.e., the ground-truth answer is wrong. This may leave the reader with the impression that this slight inaccuracy is a necessary condition. However, the paper later shows that using the correct answer "12.7" (Row 11 in Table 5) works similarly to the wrong answer "12.8" (Row 5), with the difference likely being statistically insignificant. It would be smoother if the narrative started with the correct answer and then showed that the result is robust to slight deviations. I understand this would require a substantial re-running of existing experiments, so perhaps more clarification in the writing is sufficient.
- Informal writing style: The writing tone can be more academic—for example, by using less italics and bolding. Page 8 is filled with bolded sentences, which is atypical of an academic paper. Informal phrases such as "What's more" should be removed.

Overall, the results in the paper are novel and strong, and the weaknesses above are relatively minor, so I will recommend acceptance.

---

> ### Author Rebuttal · Authors · 2025-07-31
>
> Thank you for your recognition of our paper together with your valuable comments and suggestions. We will revise our paper according to your comments. We respond to your comments below and would appreciate it if you could let us know if our response addresses your concerns.
>
>
> > **Q1**: A more in-depth analysis could strengthen the paper. Section 4.1 discusses that the underlying reason for the phenomenon is not due to grokking, while Section 4.2 shows that entropy alone may also help in the performance gain. However, neither section analyzes the underlying reason why a single sample is sufficient.
>
>
> **A1**: As you mentioned, in our paper, we mainly focus on empirical findings and demonstrate some necessary conditions for the success of one(few)-shot RLVR, such as the strong capabilities stored in the base model due to pretraining/midtraining stages (Lines 810–819), the importance of policy gradient and exploration, etc.
>
> While more theoretical analysis would certainly be valuable, it would also require significant additional effort due to the complexity of the problem, especially if a rigorous proof is to be pursued. Such analysis could become an independent contribution and is difficult to fully include within the limited space of this rebuttal. Therefore, we believe it is reasonable to leave the theoretical investigation as future work.
>
> In fact, we also discuss possible explanatory directions in Appendix D.3 (Lines 838–846), suggesting ideas for future research. For example, if a specific training example can encourage the model to output diverse Chain-of-Thoughts (CoTs) that generalize to other problems—especially when aided by entropy loss—then the policy gradient might serve as an implicit regularizer on these exploratory CoTs (e.g., punishing the wrong exploration). This could help the model leverage capabilities (e.g., diverse reasoning patterns) acquired during pretraining and midtraining. Additionally, it might be possible to draw on techniques used in analyzing double descent (note that the 1-shot RLVR on $\pi_{13}$ shows a test curve somewhat similar to “double descent,” as in Fig. 2 middle), which is related to the implicit regularization effects of SGD, to better understand the behavior of 1-shot RLVR. We will make these points more clear in the main paper.
>
>
>
>
> > **Q2**: Slightly misleading narrative in Section 3.2.1: In Section 3.2.1, it makes the observation that "The answer is not precise"—i.e., the ground-truth answer is wrong. This may leave the reader with the impression that this slight inaccuracy is a necessary condition. However, the paper later shows that using the correct answer "12.7" (Row 11 in Table 5) works similarly to the wrong answer "12.8" (Row 5), with the difference likely being statistically insignificant. It would be smoother if the narrative started with the correct answer and then showed that the result is robust to slight deviations. I understand this would require a substantial re-running of existing experiments, so perhaps more clarification in the writing is sufficient.
>
> **A2**:
> Thanks for pointing this out! We acknowledge that the current narrative could be slightly misleading. We will revise the clarification in Sec. 3.2.1 to reduce the emphasis on imprecise answers in $\pi_1$, as 1-shot RLVR using a single example with a precise answer (e.g., $\pi_{13}$, Line 868) also works well.
>
> We have discussed a related topic — the influence of random incorrect labels — in Sec. C.2.4 (Tab. 15), as an extension of the label correctness analysis in Tab. 5. However, as you mentioned, thoroughly investigating how varying degrees of label deviation affect RLVR would be an independent study requiring extensive experiments. We leave this as an interesting direction for future work.
>
>
>
> > **Q3**: Informal writing style: The writing tone can be more academic—for example, by using less italics and bolding. Page 8 is filled with bolded sentences, which is atypical of an academic paper. Informal phrases such as "What's more" should be removed.
>
>
> **A3**: Thanks for pointing this out! We will definitely revise our paper based on this feedback in the next version!
>
>
>
>
> > **Q4**: In Figure 4, I observe that the response length of training on a single sample is much longer than on the full dataset. Would the response length in the evaluation dataset also be substantially longer? If so, this may be a significant drawback of training on a single sample that needs to be emphasized, due to the increased inference cost (even if the performance is similar).
>
>
> **A4**: The response lengths in the evaluation set would **not** be substantially longer and still remain within a reasonable range. This aligns with the post-saturation generalization phenomenon described in Sec. 3.2.2, where the model begins to overfit to the single training example—mixing correct reasoning with excessively long, meaningless outputs on that example (Fig. 3, Step 1860, Left)—while still producing reasonable outputs on the test set (Fig. 3, Step 1860, Right).
>
> For reference, the average response lengths (i.e., number of output tokens) on the test tasks are shown below. While 1-shot RLVR generally produces longer responses than RLVR with DSR-sub, the difference remains within a reasonable range and is consistent with our observations regarding the frequency of reflection (Fig. 4, Right). We would add these discussions in the revised paper.
>
>
> **Table R2**: Average response length of Qwen2.5-Math-1.5B on evaluation tasks. Here we use format-reward experiment (DSR-sub + Format in Tab. 12, Sec. C.2.3) as a baseline for eliminating the difference in tokens introduced by formats.
>
> | |MATH500|AIME 2024|AMC 2023|Minerva Math|OlympiadBench|AIME 2025|Avg.
> |-|-|-|-|-|-|-|-|
> Format-Fixing|689.0|1279.7 |911.0|1018.1 |957.3|1177.1 |1005.4
> 1-shot RLVR (step 100)|611.1 |1122.7 | 939.1 |1071.8 | 951.2 |1173.0 | 978.2
> 1-shot RLVR (step 1500)|739.7 |1352.4 | 985.6 |904.7 |1088.9 | 1250.7 | 1053.7
> RLVR with DSR-sub (step 100)|635.9|1268.0| 873.7|797.0|953.6|1121.6| 941.6
> RLVR with DSR-sub (step 1500)|561.7|949.1|761.8|637.6|783.8|988.4|780.4
>
>
>
>
> > **Q5**: The evaluation mainly uses pass@1 with temp=0 and pass@1 with temp=0.6. I am interested in how pass@n (e.g., n=8, with a decent temperature such as 0.6) compares between training on a full dataset and a single sample. Given the large entropy shown in Figure 4 for 1-shot training, will pass@n be worse than when training on a full dataset?
>
>
> **A5**: This is a great question. As you suggested, we ran this analysis for AIME 24, AMC23 and AIME25 (pass@8, temperature = 0.6) on Qwen2.5-Math-1.5B, and the results are shown in Tab. R3 below:
>
> **Table R3**: Pass@8 results on 3 math evaluation tasks using Qwen2.5-Math-1.5B. We also include the performance of RLVR with format-reward (as in Table R2) as a stronger baseline.
>
> | |AIME 2024|AIME 2025|AMC 2023|Avg on 3 tasks
> |-|-|-|-|-|
> Base Model|26.6|20.0|72.5|39.7|
> Format-Fixing (highest) |33.3|23.3|72.5|43.1|
> |
> RLVR with DSR-sub (highest, step 160) | 36.7| 26.7 | 87.5 | 50.3 |
> RLVR with DSR-sub (step 500) | 33.3|30.0|82.5|48.6|
> RLVR with DSR-sub (step 1000) | 33.3|20.0|75.0|42.8|
> RLVR with DSR-sub (step 1500) | 30.0|26.7|67.5|41.3|
> |
> 1-shot RLVR (step 500)| 30.0|16.7|80.0|42.2|
> **1-shot RLVR (highest, step 980)**| 36.7 | 33.3 | 85.0 | **51.7** |
> 1-shot RLVR (step 1500)| 26.6|23.3|87.5|45.8|
>
>
> Interestingly, we find that
>
> (1) 1-shot RLVR achieves similar or even slightly better best pass@8 performance (51.7%) across these three math tasks compared to full-set RLVR (50.3%). These results show that 1-shot RLVR will not result in worse pass@n performance, and also supports our claim about post-saturation generalization (Fig. 3): even though the model overfits to the single training example and shows high entropy on it (Fig. 4 middle), it still performs well on evaluation tasks.
>
> (2) There is a noticeable downward trend in pass@8 performance for full-set RLVR (with 1.2k DSR-sub) after 200 steps. This phenomenon is consistent with recent findings that RLVR can sometimes hurt a model’s pass@n performance [1].
>
>
> Thanks for your insightful suggestion. We will include these new findings in the next revised version.
>
>
>
> [1] Yue, Yang, et al. "Does reinforcement learning really incentivize reasoning capacity in llms beyond the base model?." arXiv preprint arXiv:2504.13837 (2025).

---

### Official Review · Reviewer_U7dr · 2025-06-30

**Clarity:** 3
**Significance:** 2
**Originality:** 2
**Rating:** 3
**Confidence:** 4

**Summary:**

This paper finds that RL training on only one example can improve the reasoning performance of LLM.
The paper explores the effectiveness of different training examples and finds that they can all somehow improve the performance. The performance on the test set also increases as the training becomes longer, even if the accuracy on the training sample saturates.

**Questions:**

See weakness

**Ethical Concerns:**

["NO or VERY MINOR ethics concerns only"]

**Final Justification:**

It seems all the reviewers point out that more depth analysis is necessary. However, the author response does not resolve our concerns at this point. We agree that the phenomenon that training on only one of some examples could increase the performance is interesting; however, it is merely a discovery, as a scientific research project, only pointing out the phenomenon without solid explanation/proof/proposed method is incomplete.
The reviewer would expect more solid explanations or an effective method that exploits this phenomenon in this paper. Therefore, I believe this paper is not fully ready for this venue, although it is interesting.

**Limitations:**

See weakness.

**Quality:**

3

**Strengths And Weaknesses:**

## Strengths
1. The paper reveals an interesting phenomenon, that RLVR with only one example would improve the reasoning performance of LLM. The paper further explores different training samples, and the phenomena on training/test set during training.

## Weaknesses
1. The paper mainly presents empirical experimental results with hypotheses, but does not discuss why or provide convincing explanations with solid support.
2. The training setting seems to be rarely used during real training, which makes the conclusion less applicable.
3. The proposed method for selecting examples does not work, since training with samples $\pi_{13}$ and $\pi_{1209}$ also obtains decent improvements.
4. The policy gradient loss is the basic functional part in RLVR. The reviewer does not quite understand the meaning of Rows 1-10 in Table 5. It is obvious that the policy gradient loss dominates the learning from the reasoning trajectories. The reviewer also does not understand the logic in line 258/259, why 1-shot RLVR is attributed to policy gradient loss can distinguish it from “grokking”.
5. What does 'post-saturation generalization' mean? We might need further explanation. From Figure 2, it seems that training on only one example cannot surpass the best performance of training on many examples, and it seems training on many samples brings faster growth of accuracy on the test set, so what is the advantage of training on one example?

---

> ### Author Rebuttal · Authors · 2025-07-31
>
> Thank you for your recognition of our paper and your valuable feedback. We have responded to your concerns and will revise our paper based on the discussions. We would also appreciate it if you could let us know if our response addresses your concerns.
>
>
>
> > **Q1**: The paper mainly presents empirical experimental results with hypotheses, but does not discuss why or provide convincing explanations with solid support.
>
> **A1**:
>
> In our paper, we focus on empirical findings and demonstrate some necessary conditions for the success of one(few)-shot RLVR, such as the strong capabilities stored in the base model due to pretraining/midtraining stages (Lines 810–819), the importance of policy gradient and exploration (but not weight decay, unlike grokking, see Sec. 4.1), etc. We have conducted extensive experiments to confirm and illustrate the key features of one(few)-shot RLVR (Sec. 3, Sec. 4, Appendix C).
>
> While more theoretical analysis would certainly be valuable, it would also require significant additional effort due to the complexity of the problem, especially if a rigorous proof is to be pursued. Such analysis could become an independent contribution and is difficult to fully include within the limited space of this rebuttal. Therefore, we believe it is reasonable to leave the theoretical investigation as future work.
>
> In fact, we also discuss possible explanatory directions in Appendix D.3 (Lines 838–846), suggesting ideas for future research. For example, if a specific training example can encourage the model to output diverse Chain-of-Thoughts (CoTs) that generalize to other problems—especially when aided by entropy loss—then the policy gradient might serve as an implicit regularizer on these exploratory CoTs (e.g., punishing the wrong exploration). This could help the model leverage capabilities (e.g., diverse reasoning patterns) acquired during pretraining and midtraining. Additionally, it might be possible to draw on techniques used in analyzing double descent (note that the 1-shot RLVR on $\pi_{13}$ shows a test curve somewhat similar to “double descent,” as in Fig. 2 middle), which is related to the implicit regularization effects of SGD, to better understand the behavior of 1-shot RLVR. We will make these points more clear in the main paper.
>
>
>
>
>
>
>
> > **Q2**: The training setting seems to be rarely used during real training, which makes the conclusion less applicable.
>
>
> **A2**: We would like to address this concern from the following points:
>
> (1) Few-shot reinforcement learning for LLMs remains valuable in **data-limited** domains where obtaining large amounts of high-quality, verifiable data is difficult and often expensive to annotate. For example, OpenAI’s reinforcement fine-tuning also supports training models with as few as 10 examples in fields like medicine [3]. Besides, we think one(few)-shot learning will also be helpful for other data-centric case studies in RLVR, because we can observe how each single data point or small amount of data affects the RLVR process and model output styles.
>
> (2) Most importantly, the main target of our paper is analyzing a new observation of RLVR, rather than proposing a new method. Except for the dataset size, all other experimental setups are very similar to other RLVR studies. And we think it conveys or confirms a lot of messages applicable for RLVR with general datasets, such as:
>
> (a) Supporting that the base model already has strong capability (discussed in Line 810–819), because one example brings limited new information and instead incentivizes the LLM’s pre-learned capabilities — emphasizing the importance of pretraining and midtraining for RLVR.
>
> (b) Showing that RLVR can be very hard to overfit to the data (as we train on 1 example thousands of times). This is a nice signal for training stability.
>
> (c) Showing that RLVR can be data-efficient, but still compute-intensive. The amount of RLVR data used in current RLVR recipes is much less than in other stages (e.g., SFT or midtraining). This implies the importance of data selection and curation for RLVR (discussed in D.3 Line 830–837). For example, Tulu3 [2] uses 939k examples for SFT, while Qwen3 [1] uses only 4k query-verifier pairs for reasoning RL.
>
> (d) Supporting the importance of encouraging exploration in the RLVR process (discussed in Line 847–858). In Fig. 5 we show that adding entropy loss is important for the post-saturation stage in 1-shot RLVR, and slightly increasing training temperature (e.g., from 0.6 to 1.0) can further improve results. Better exploration strategies are also important for future RLVR training.
>
> We think these topics are important for real RLVR training and believe our findings would be insightful for future study. We will add these discussions in the revised paper to better clarify our position.
>
>
>
>
>
> > **Q3**: The proposed method for selecting examples does not work, since training with samples pi_13 and pi_1209 also obtains decent improvements.
>
>
>
> **A3**: We would reply to this question in the following points:
>
> (1) **Our selection method still outperforms random baselines.** In Table 4 (Qwen2.5-Math-7B), our selected set $\{\pi_1, ..., \pi_{16}\}$ achieves 42.5% accuracy—close to the 42.8% from full 1.2k DSR-sub RLVR, and clearly better than 40.2% from randomly selected examples.
>
> (2) We apply historical variance score for filtering examples with critical errors in ground truth or overly difficult problems (e.g., $\pi_{1207}$ and $\pi_{1208}$), which are not suitable for few-shot RLVR (Tab. 3). Random selection risks including such examples.
>
> (3) Most importantly, our paper focuses on analyzing a new feature of RLVR—not on proposing a SOTA data selection method (Lines 121–125). In fact, the success of $\pi_{13}$ and $\pi_{1207}$ **supports the generality** of 1-shot RLVR, rather than undermining our selection strategy. While better selection techniques are valuable, they are beyond our scope.
>
>
>
> > **Q4-1**: The reviewer also does not understand the logic in line 258/259, why 1-shot RLVR is attributed to policy gradient loss can distinguish it from “grokking”.
>
> **A4-1**:
> “Grokking” is a phenomenon that is strongly affected by explicit regularization methods like weight decay or dropout, as shown in the original grokking paper [4] and mentioned in Lines 241–248 of our main paper. However, from Row 2 vs. Row 3 or Row 8 vs. Row 4 in Tab. 5 (or Fig. 14), we can see that weight decay has little effect in 1-shot RLVR training. Therefore, 1-shot RLVR are different from "grokking".
>
>
>
> > **Q4-2**: The policy gradient loss is the basic functional part in RLVR. The reviewer does not quite understand the meaning of Rows 1-10 in Table 5. It is obvious that the policy gradient loss dominates the learning from the reasoning trajectories.
>
>
> **A4-2**:
> We respectfully disagree with the question. There are four loss terms influencing RLVR training, and both weight decay and policy gradient could contribute to the observed post-saturation phenomenon. To draw firm conclusions, we believe the ablations in Tab. 5 are necessary.
>
> As discussed in A4-1 and Lines 241–248, the post-saturation generalization shows similarities to the “grokking” phenomenon, which is known to be strongly affected by weight decay. However, we cannot conclude whether the success of 1-shot RLVR is due solely to weight decay, solely to policy gradient, a combination of both, or interactions with other terms such as entropy loss—without conducting proper ablation studies.
>
> While the policy gradient loss is indeed crucial for RLVR training, it is not necessarily the primary reason for the new phenomenon. For example, in the “grokking” paper [1], the training loss is critical, but regularization techniques like weight decay have a significant impact on the final outcome.
>
> Additionally, we observed a ~4% improvement on MATH500 when combining entropy loss and policy gradient (Row 7) compared to using policy gradient alone (Row 2) in 1-shot RLVR. This supports the value of exploration (Fig. 5) and reinforces the need for our full ablation analysis.
>
>
>
>
> > **Q5-1**: What does 'post-saturation generalization' mean? We might need further explanation.
>
>
> **A5-1**: “Post-saturation generalization” refers to the phenomenon in 1-shot RLVR where, after the model reaches near 100% accuracy on the training data, its test performance continues to improve as training progresses.
> We mention this in the title of Sec. 3.2.2, the caption of Fig. 2, and Lines 174–178. We could provide a more formal definition in the main paper.
>
>
>
>
>
> > **Q5-2**:... training on only one example cannot surpass the best performance of training on many examples, and it seems training on many samples brings faster growth of accuracy on the test set, what is the advantage of training on one example?
>
>
> **A5-2**: The main focus of our paper is to analyze a new observation in RLVR that we believe will be insightful for future research and real-world applications, rather than to directly propose a new training method. This response largely overlaps with A2 above.
>
> In addition, although one(few)-shot RLVR indeed requires more computation to converge, it significantly improves data efficiency in RLVR training (using less than 0.1% of the data), which is particularly valuable for data-limited domains (refer to A2), given the high cost of annotating high-quality RLVR data.
>
>
>
> ---
> [1] Yang, An, et al. "Qwen3 technical report." arXiv preprint arXiv:2505.09388 (2025).
>
> [2] Lambert, Nathan, et al. "Tulu 3: Pushing frontiers in open language model post-training." arXiv preprint arXiv:2411.15124 (2024).
>
> [3] OpenAI, “Reinforcement Fine‑Tuning,” OpenAI Platform Documentation, 2025. [Online]. Available: https://platform.openai.com/docs/guides/reinforcement-fine-tuning. [Accessed: Jul. 30, 2025].
>
> [4] Power, Alethea, et al. "Grokking: Generalization beyond overfitting on small algorithmic datasets." arXiv preprint arXiv:2201.02177 (2022).

---

> > ### Author Response · Authors · 2025-08-07
> >
> > We would like to sincerely thank you once again for your valuable time and feedback on our paper! We believe our rebuttal has addressed your concerns. As we are approaching the final day of the review process, we kindly ask if you had a chance to review our responses.  Please feel free to let us know if there is anything else we can clarify or discuss, thanks!

---

### Official Review · Reviewer_bR1Q · 2025-07-02

**Clarity:** 3
**Significance:** 3
**Originality:** 3
**Rating:** 5
**Confidence:** 4

**Summary:**

This paper demonstrates that modern large language models can be fine-tuned via reinforcement learning on a single automatically verifiable training example to achieve dramatic improvements in mathematical reasoning: for instance, lifting accuracy on the MATH500 benchmark from 36 % to 73.6 %—on par with methods using thousands of examples. Through extensive experiments across multiple base models (Qwen, Llama, DeepSeek), RL algorithms (PPO, GRPO), and ablations, the authors show that both reward-based and entropy-only objectives yield large gains, that these gains transfer across problem domains, and that test performance continues to improve long after the model perfectly fits the lone example. This “one-shot” RL approach challenges prevailing assumptions about data requirements for RL tuning in LLMs and opens new avenues for data-efficient, exploration-driven fine-tuning.

**Questions:**

I greatly enjoyed the paper and find the empirical findings persuasive. To help researchers and practitioners translate these insights into day-to-day RLVR workflows, could the authors elaborate on a few near-term, “actionable” takeaways? This is a general question, which I wish to learn from the authors.

**Ethical Concerns:**

["NO or VERY MINOR ethics concerns only"]

**Final Justification:**

The authors have answered my question properly and I believe the contribution of this work is important for understanding RLVR in LLM.

**Limitations:**

Yes

**Quality:**

3

**Strengths And Weaknesses:**

Strengths
- Timely focus on data efficiency – Tackles the under-explored but important question of how little data RL-based verification–reward pipelines really need, pushing the field toward more resource-friendly methods.
- Non-trivial findings – Demonstrates that training on just one verifiable example can unlock large gains, an observation that is both surprising and likely to inspire follow-up work.
- Thorough experimental coverage – Evaluates multiple base models, RL algorithms, training recipes, and a wide range of ablations, giving the results credibility and making the paper a useful reference point.

Weaknesses
- Limited explanatory depth – While the empirical phenomena are intriguing, the paper offers only light, largely speculative explanations; a clearer theoretical or mechanistic account would strengthen the contribution, which I understand is a bit too much to ask for but would significantly strengthen the contribution.

---

> ### Author Rebuttal · Authors · 2025-07-31
>
> Thanks a lot for your recognition of our paper together with your valuable comments and suggestions. We respond to your questions below and would appreciate it if you could let us know if our response addresses your concerns.
>
>
>
> > **Q1**: Limited explanatory depth – While the empirical phenomena are intriguing, the paper offers only light, largely speculative explanations; a clearer theoretical or mechanistic account would strengthen the contribution, which I understand is a bit too much to ask for but would significantly strengthen the contribution.
>
>
> **A1**: As you mentioned, in our paper, we mainly focus on empirical findings and demonstrate some necessary conditions for the success of one(few)-shot RLVR, such as the strong capabilities stored in the base model due to pretraining/midtraining stages (Lines 810–819), the importance of policy gradient and exploration, etc.
>
> While more theoretical analysis would certainly be valuable, it would also require significant additional effort due to the complexity of the problem, especially if a rigorous proof is to be pursued. Such analysis could become an independent contribution and is difficult to fully include within the limited space of this rebuttal. Therefore, we believe it is reasonable to leave the theoretical investigation as future work.
>
> In fact, we also discuss possible explanatory directions in Appendix D.3 (Lines 838–846), suggesting ideas for future research. For example, if a specific training example can encourage the model to output diverse Chain-of-Thoughts (CoTs) that generalize to other problems—especially when aided by entropy loss—then the policy gradient might serve as an implicit regularizer on these exploratory CoTs (e.g., punishing the wrong exploration). This could help the model leverage capabilities (e.g., diverse reasoning patterns) acquired during pretraining and midtraining. Additionally, it might be possible to draw on techniques used in analyzing double descent (note that the 1-shot RLVR on $\pi_{13}$ shows a test curve somewhat similar to “double descent,” as in Fig. 2 middle), which is related to the implicit regularization effects of SGD, to better understand the behavior of 1-shot RLVR. We will make these points more clear in the main paper.
>
>
>
> > **Q2**: I greatly enjoyed the paper and find the empirical findings persuasive. To help researchers and practitioners translate these insights into day-to-day RLVR workflows, could the authors elaborate on a few near-term, “actionable” takeaways? This is a general question, which I wish to learn from the authors.
>
>
> **A2**: Thanks a lot for your recognition of our work! We are very glad to discuss our thoughts on RLVR with you.
>
> (1) **Design better exploration strategies**. In Sec. 4.1, we show that adding entropy loss and properly increasing the sampling temperature can improve the performance of one(few)-shot RLVR. We believe exploration is one of the key factors in RLVR. The current entropy loss evaluates token-level diversity, which may not be the best metric for exploration. Alternative metrics like semantic entropy or sentence-level embedding diversity could potentially perform better. It would also be interesting to explore more complex and effective exploration strategies beyond standard parallel rollouts.
>
> (2) **Investigate mid-training and RLVR jointly, and explore more domains**. The effectiveness of 1-shot RLVR suggests that the model already possesses strong capabilities from the pretraining or mid-training stages. Therefore, a more comprehensive investigation of RLVR could involve jointly controlling both mid-training and RLVR data. Since the math domain is often fully covered during mid-training, it would be valuable to apply comprehensive investigation to more diverse domains.
>
> (3) **Move from data efficiency to compute efficiency**. As shown in the paper, RLVR is highly data-efficient but computationally expensive. Given a strong base model, it is meaningful to explore how to better leverage its capabilities with minimal compute. For instance, in Tab. 14 of Appendix C.2.3, we interestingly find that using $\pi_1$ and the model’s own CoT output (from step 500) as an in-context example can significantly improve Qwen2.5-Math-7B’s performance (37.4%, compared to 34.3% for the format-fixing baseline). This single example significantly outperforms the 4 official examples provided by the Qwen2.5 repository (21.7%). Although it does not work perfect on Qwen2.5-Math-1.5B, it is interesting to further explore such lightweight or zero-train methods for boosting performance.
>
> (4) **Better understand post-saturation generalization from the data perspective**. In Appendix C.2.5, we observe that simplifying the problem in $\pi_1$ significantly reduces its effectiveness in 1-shot RLVR (Tab. 16), highlighting the importance of the problem statement and what kind of CoTs it can incentivize. It is also noteworthy that different data points vary widely in their impact on 1-shot RLVR performance (Tab. 3). A promising future direction is to use a similar 1-shot RLVR process to study how individual problems affect CoT generation and model behavior, which may also provide insight into post-saturation generalization and help with data curation.
>
> We include some related discussions in Appendix D.3 and believe our findings will be helpful for future work.

---

> > ### Comment · Reviewer_bR1Q · 2025-08-04
> >
> > The authors' comments are really appreciated! I would like to maintain my favorable consideration of this work.

---

> > > ### Author Response · Authors · 2025-08-04
> > >
> > > Thanks again for your suggestion and the recognition to our work!

---

### Official Review · Reviewer_yG1q · 2025-07-03

**Clarity:** 2
**Significance:** 4
**Originality:** 3
**Rating:** 5
**Confidence:** 4

**Summary:**

This paper studies the importance of training set size for GRPO training with the goal of improving reasoning capabilities of the models. Interestingly, the paper shows that there are individual samples (in training sets) where training with that individual sample alone would almost reach the same performance as training with the whole dataset. Authors attribute this observation partly to the increase in self reflective behaviors of the model even when a sample is learned.

**Questions:**

- 1. In GRPO training, if all rollouts receive reward 1 there will be no update from the policy gradient which happens to be the case after the first 200 iterations. So I wonder maybe we can get the same results with a combination of fixing the format (probably the first 100 iterations with 1 or more samples) and then just entropy training? (I.e., there is nothing coming from the policy update in post-saturation right?)
- 2. Why self reflection doesn’t increase when we train on the full data? How can one boost that?
- 3. What response length did you use for the 7B and 3B models?
- 4. Do you think there will be a way to select the data without training first for E epochs? In the paper do you use the historical variance score of Qwen2.5-Math-1.5B for all the models?
- 5. Can you elaborate on "To enable RLVR with one or few examples, we duplicate the chosen data until they reach the training batch size (e.g., 128) and store them as a new dataset."?
- 6. How do you report accuracies for the experiments, because the accuracy is quite unstable across different iterations?

**Ethical Concerns:**

["NO or VERY MINOR ethics concerns only"]

**Final Justification:**

I also believe this paper provides important insights into the role of RL in increasing the reasoning performance of language models. We do not yet fully understand, how one sample can be that effective. I also believe there might be scenarios (datasets + models) in which full RL training is required. Nevertheless, I think the community would benefit from this paper.

**Limitations:**

discussed in the paper

**Paper Formatting Concerns:**

None.

**Quality:**

3

**Strengths And Weaknesses:**

## Strengths
The paper is full of interesting observations and reports. For instance there is post-saturation generalization, even when a model achieves perfect accuracy on a single training sample, further training on the sample would still improve test performance. Subsequently, the paper shows the importance of the entropy term in boosting the performance of models.
## Weaknesses
- The explanations provided for the current observations are rather preliminary and there are still many unanswered questions (many of them may become future research questions, see below for some).
- As authors have pointed out, a huge part of the gains are only due to format fixing. The gains due to format fixing may cause misinterpretation in the results of the paper. Here are some details. In Table 12, we observe that format fixing can increase model's performance to ~29% on the average score and to 65% on MATH. In Table 3, the 'performance increase' of several individual samples have been reported. We see that for samples {4, 7, 11} the performance reaches 65% after GRPO training and this is reported as 28% increase in performance. However, if we just compare this with format fixing, we can conclude these trainings are actually not doing anything beyond format fixing, but unfortunately the current presentation could be misleading. There is also another subtle phenomenon. The reader may compare performance increase (for MATH dataset) with the full set (39.4%) and the performance increase from one sample (38%) and concludes that they are almost the same. However, if we consider the baseline that only fixes the format (which already has 29% performance increase), we reach performance increase of 10.4% for the full training set vs. 9% for an individual sample. The point is there's much more difference between 10.4% and 9% compared to 39.4% and 38%. This baseline issue goes for most tables and figures presented in the table.


I'd like to be extra clear. I think the paper's results are still significant and very interesting. However, I think they are slightly misleading in the current form, hence I will be happy to increase my score if the authors take proper actions to fix this issue. (One suggestion could be including format-fixing training as a baseline in tables and figures).

---

> ### Author Rebuttal · Authors · 2025-07-31
>
> Thank you for your recognition of our work and your constructive feedback to improve our paper. We will revise our paper based on your advice. We detail our response below and please kindly let us know if our response addresses your concerns.
>
> > **Q1**: The explanations provided for the current observations are rather preliminary and there are still many unanswered questions
>
> **A1**:
> As you mentioned, in our paper, we mainly focus on empirical findings.
> While more theoretical explanations would certainly be valuable, it would also require significant additional effort due to the complexity of the problem, especially if a rigorous proof is to be pursued. Such analysis could become an independent contribution and is difficult to fully include within the limited space of this rebuttal. Therefore, we believe it is reasonable to leave the theoretical investigation as future work.
>
> In fact, we also discuss possible explanatory directions in Appendix D.3 (Lines 838–846), suggesting ideas for future research. For example, if a specific training example can encourage the model to output diverse Chain-of-Thoughts (CoTs) that generalize to other problems—especially when aided by entropy loss—then the policy gradient might serve as an implicit regularizer on these exploratory CoTs (e.g., punishing the wrong exploration). This could help the model leverage capabilities (e.g., diverse reasoning patterns) acquired during pretraining and midtraining. Additionally, it might be possible to draw on techniques used in analyzing double descent (note that the 1-shot RLVR on $\pi_{13}$ shows a test curve somewhat similar to “double descent,” as in Fig. 2 middle), which is related to the implicit regularization effects of SGD, to better understand the behavior of 1-shot RLVR. We will make these points more clear in the main paper.
>
> > **Q2**: As authors have pointed out, a huge part of the gains are only due to format fixing… One suggestion could be including format-fixing training as a baseline in tables and figures
>
> **A2**: Thank you very much for bringing this up! We fully agree with your suggestion to prevent potential misinterpretation of format-fixing gains. To better disentangle the effects of format-fixing and non-format improvements, we plan to revise the main paper to include additional clarifications and baselines. Specifically, we will at least consider:
>
> - More clearly distinguish format and non-format improvements in abstract and introduction.
> - Add a baseline to Fig. 1, 6 and Tab. 3, 5, and 6, along with corresponding discussions. For example, as you suggested, we will revise the improvement values in Table 3 and show that some data do not yield improvements beyond format-fixing.
> - Include baselines in Table 4 for 1.5B and 7B. We omit Llama-3.2-3B-Instruct and DeepSeek-R1-Distill-Qwen-1.5B, as these models already perform well with the default chat template due to instruction tuning and distillation. The format-reward baseline for Qwen2.5-Math-1.5B is in Sec. C.2.3, and that for Qwen2.5-Math-7B (new results) is as below.
>
> **Table R1**: RLVR with format-reward for Qwen2.5-Math-7B.
> | |Dataset Size|MATH500|AIME24|AMC23|Minerva Math|OlympiadBench|AIME25 | Avg.
> |-|-|-|-|-|-|-|-|-|
> Base  | 0 | 51.0 | 12.1 | 35.3 | 11.0 | 18.2 | 6.7 | 22.4 |
> +format-reward | 1209 | 65.8 | 24.2 | 54.4 | 24.3 | 30.4 | 6.7 | 34.3 |
> **+1-shot RLVR** | 1 | 79.2 | 23.8 | 60.3 | 27.9 | 39.1 | 10.8 | 40.2 |
>
> We will incorporate the above changes in the final version of the paper.
>
>
>
> > **Q3**: if all rollouts receive reward 1 there will be no update from the policy gradient which happens to be the case after the first 200 iterations. we can get the same results with a combination of fixing the format (probably the first 100 iterations with 1 or more samples) and then just entropy training? (I.e., there is nothing coming from the policy update in post-saturation right?)
>
>
> **A3**: The rollout accuracy after 200 training steps is very close to 100%, but importantly, still slightly below it (e.g., around 99%). This means that incorrect rollouts can still have a large advantage due to small variance (as Eq. 6 in Appendix B.1), while lots of correct rollouts gets a minor but non-zero advantage. As a result, the policy gradient updates remain important after 200 steps, as the gradient norm shown in Fig. 18. Related discussions are in Appendix D.2.1.
>
> Actually, policy gradients are essential to penalize outputs that fail to yield correct answers. Without this, training with entropy loss alone quickly leads to excessive entropy and degenerate outputs (As shown in Fig. 14, gets 0 accuracy after 20 steps). Similarly, in Tab. 6 and Appendix C.2.2 (Lines 709–715), we mention that entropy-only training is only effective for the initial few steps. Therefore, we believe combining only format reward and entropy training is insufficient. Without policy gradients, the model may degenerate into generating random outputs containing “\boxed{}”, lacking meaningful CoTs. We will clarify these more explicitly in the main paper.
>
>
>
> > **Q4**: Why self reflection doesn’t increase when we train on the full data? How can one boost that?
>
>
> **A4**: Thanks for the good question. We think this phenomenon likely stems from two factors:
> (1) the max training response length, and
> (2) varying response lengths across data (often tied to problem difficulty).
>
> A good illustration of this can be found in DeepScaleR [3] (Fig. 2).
> When the maximum training context length is limited (e.g., 8k), excessive self-reflection may lead to overly long responses that are eventually clipped. Additionally, GRPO loss tends to favor shorter correct responses over longer ones [2]. So model may compress its CoT, reducing response length and self-reflection.
> In contrast, longer context windows (e.g., 16K in DeepScaleR) enable effective but not excessively long self-reflections on more difficult problems—the model has more room to reason and can obtain higher rewards. In this case, the average response length tends to increase.
>
> In our case, Qwen2.5-Math-1.5B has a 4096-token context limit, and with a 1024-token prompt, responses are capped at 3072 tokens. This constraint likely contributes to the drop in both response length and reflection. But generally, when using models with longer context windows, adjusting max training response length and data difficulty may better support self-reflection on challenging tasks.
>
>
>
> > **Q5**: What response length did you use for the 7B and 3B models?
>
> **A5**: For both Qwen2.5-Math-7B and Llama-3.2-3B-Instruct, we use a response length of 3072 for both training and evaluation (Train: Line 144–145, Eval: Line 628).
>
> > **Q6**: Do you think there will be a way to select the data without training first for E epochs? In the paper do you use the historical variance score of Qwen2.5-Math-1.5B for all the models?
>
> **A6**: Yes, for simplicity, we use the same historical variance score from Qwen2.5-Math-1.5B to select data for all models. Although this score currently requires training models before compute, developing more efficient data selection or scheduling methods is an interesting future direction (Appendix D.3, Line 830-837). Potential approaches include:
>
> 1. **Proxy model**: Similar to us, use a smaller but appropriate proxy model to compute the historical variance score,and then be applied to larger models. This is more efficient than training each target model individually.
>
> 2. **Learnability-based metrics[1]**: For example, one can select a suitable teacher model and compute the accuracy gap between the teacher and target models for each problem. This gap serves as a proxy for learnability and can be used for filtering.  This approach only requires a few inference passes over the training data, and we have found it performs reasonably well in real-world RLVR scenarios.
>
> Although designing a new data selection method is not the main focus of our paper, it would be valuable to explore whether historical variance score can be combined with efficient selection techniques. The historical variance score may also be useful for data scheduling in RLVR, where training data is scheduled based on prior training steps, potentially avoiding additional costs when calculating the score.
>
> > **Q7**: elaborate "To enable RLVR with one or few examples, we duplicate the chosen data until they reach the training batch size (e.g., 128) and store them as a new dataset."
>
> **A7**: This is a technical choice that doesn't affect our conclusions. For consistency, we use the same batch size (128) in 1-shot RLVR as in full RLVR with DSR-sub. Since `verl` sets `drop_last=True` for the training dataloader, the dataset must be at least as large as the batch size; otherwise, all data would be dropped. So we duplicate the selected examples until the dataset reaches size 128. We will clarify this in the revised paper.
>
> > **Q8**: How do you report accuracies for the experiments, because the accuracy is quite unstable across different iterations?
>
> **A8**: By default, in main Tables/Figures (e.g., Fig. 1, 6; Tab. 4, 6, 7), we report the checkpoint with the best average performance across 6 math benchmarks (saved every 20 steps). For comparisons or ablations on only MATH500 and AIME24 (e.g., Tab. 3, 5), we report the best performance on each benchmark separately. These details are noted in the appendix (Lines 633–638) and captions of Tab. 3 and 5. All methods, including baselines, follow the same setting for fairness.
>
> [1] Mindermann, Sören, et al. "Prioritized training on points that are learnable, worth learning, and not yet learnt." International Conference on Machine Learning. PMLR, 2022.
>
> [2]  Liu, Z., Chen, C., Li, W., Qi, P., Pang, T., Du, C., Lee, W. S., and Lin, M. Understanding r1-zero-like training: A critical perspective, 2025b. URL https://arxiv.org/abs/2503.20783.
>
> [3] Luo, Michael, et al. "Deepscaler: Surpassing o1-preview with a 1.5 b model by scaling rl." Notion Blog (2025).

---

> > ### Comment · Reviewer_yG1q · 2025-08-03
> > **Thanks**
> >
> > I thank the authors for their rebuttal and especially for their efforts towards clarifying the way the results are reported. I will update my evaluation accordingly.

---

> > > ### Author Response · Authors · 2025-08-04
> > >
> > > Thanks again for your thoughtful suggestions that improves our paper, and we greatly appreciate your willingness to update your evaluation!

---

### Note · Authors · 2025-08-15

We sincerely thank all reviewers for their insightful feedback and constructive comments. We would like to summarize the strengths of our work highlighted by reviewers:

- **Novel and significant findings** (yG1q, bR1Q, U7dr, URox): Our demonstration shows that RLVR can get improvement by using only one training example, which is likely to inspire follow-up works.
- **Extensive experimental validation** (bR1Q, URox): Comprehensive experiments across multiple models, RL algorithms, and thorough ablation studies provide strong empirical support.
- **Timely focus on data efficiency** (bR1Q): Addresses the important but under-explored question of minimal data requirements for RLVR pipelines.
- **Interesting empirical phenomena** (yG1q, bR1Q): Post-saturation generalization, cross-domain transfer, and the importance of entropy loss offer valuable insights.

We believe our revisions have successfully addressed all major concerns raised:
- **Format-fixing baseline** (yG1q): We committed to adding format-fixing baselines to main figures/tables and clarifying the distinction between format and non-format improvements.
- **Theoretical analysis** (yG1q, bR1Q, U7dr, URox): While acknowledging that rigorous theoretical analysis would require substantial additional work and it's reasonable to leave it for future work, we provided possible directions in Appendix and clarified the necessary conditions we identified in the ablation study.
- **Training setting applicability** (U7dr): We demonstrated that our findings have broader implications for general RLVR training, including insights about base model capabilities, training stability, data efficiency vs. compute intensity, and exploration importance. We also show that our data selection method indeed works and our work is valuable for data-limited domains.
- **Explanation of terminology and ablation study** (U7dr): We explained the definition of "post-saturation generalization" and "grokking", and showed the importance of ablation study for rigorousness.
- **Writing improvements** (URox): We committed to adopting a more academic tone and improving narrative flow.
- **Test performance** (URox): We provided comprehensive responses to questions about response length, pass@n performance with detailed experimental results.

We sincerely thank reviewer bR1Q, URox, and yG1q for their positive consideration, and the willingness of yG1q to update evaluation. We also believe that we have addressed the concerns from reviewer U7dr.

---

### Decision · Program_Chairs · 2025-09-17

**Decision:**

Accept (poster)

**Comment:**

The reviewer finds this paper to present a set of significant and interesting observations regarding GRPO training, highlighting valuable findings such as post-saturation generalization and the surprising effectiveness of individual training samples. While the core results are deemed strong, the reviewer raises a crucial and valid concern that the current presentation of these results is potentially misleading. The primary weakness identified is the failure to properly baseline against the substantial performance gains, which may misrepresent the comparative performance of single-sample versus full-dataset training. However, this is an addressable issue, and the reviewers' consensus is to recommended acceptance. The authors are still advised to revising their results, tables, and figures to incorporate a more appropriate baseline comparisons that properly contextualize their findings in the final paper.